# Evaluating factors contributing to the failure of information system in the banking industry

**Syed Mithun Ali**[1]*, **S. M. Nazmul Hoq**[1], **A. B. M. Mainul Bari**[1], **Golam Kabir**[2], **Sanjoy Kumar Paul**[3]

1 Department of Industrial and Production Engineering, Bangladesh University of Engineering and Technology, Dhaka, Bangladesh, 2 Industrial Systems Engineering, University of Regina, Regina, SK, Canada, 3 UTS Business School, University of Technology Sydney, Sydney, Australia

* mithun@ipe.buet.ac.bd

## Abstract

The increasing use of Information Technology (IT) has led to many security and other related failures in the banks and other financial institutions in Bangladesh. In this paper, we investigated the factors contributing to the failurein the IT system of the banking industry in Bangladesh. Based on the experts' opinions and weight on the specified evaluating criteria, an empirical test was conducted using a rough set theory to produce a framework for the IT system failure factors. In this study, an extended approach involving the integration of rough set theory based flexible Failure Mode and Effect Analysis (FMEA) and the Technique for Order of Preference by Similarity to Ideal Solution (TOPSIS) has beenapplied to help the managers of the corresponding field to identify the factors responsible for the failure of the IT system in the banking industries and then prioritize them accordingly, for the ease of decision-making.In this research, eleven such failure factors were identified, which were then quantitatively analyzed to facilitate managers in crucial decision-making. It was observed that cyber-attack, database hack risks, server failure, network interruption, broadcast data error, and virus effect were the most significant factors for the failure of the IT system. The framework developed in this research can be utilized to assist in efficient decision-makingin other serviceindustries where IT systems play a key role. To the best of the knowledge, this is the first study thatempirically tested key failure factors of the IT system for the banking sector using an integrated method.

## 1 Introduction

The economy is the driving force of the modern world, whereinformation technology (IT) or information system (IS) plays a vital role [1]. The incorporation of technology to formulate business is no longer a new concept [2]. As a matter of fact, in this modern age, the financial market and the banking industry largely depend on IT. Research shows that cost-effective banking transactions can be conducted through e-channel only [3]. With the growth of modern IT systems and their proper utilization, the traditional banking system has experienced radical changes. For instance, banks and other Non-Bank Financial Institutions (NBFI) have

**Competing interests:** NO authors have competing interests

gone through a paradigm shift over the past few years. IT systems have significantly improved the banking business, and have gradually made the business dependent on itself [4]. Moreover, IT plays the lead role in the digitalization of the banking systems, meeting market demands, and maintaining healthy competition with the competitors [5]. As per the World Bank report (released in August 2020), the global economy is positively being transformed by the rise in the adoption of digital business models [6] and the usage of digital financial services.

In recent times, the banks and other financial industries are adopting more and more new technologies in their businesses, to streamline their operations and to gain significant advantages in the increasingly competitive market [7,8]. Consequently, there has been a drastic shift in the way that modern customers now access their financial services. An increasing number of customers are now using digital or IT financial services via computers or mobile devices. As customers come to rely more heavily on these IT channels, the resilience and availability of these channels have become an important issue, since it is likely that even any brief disruption in these channels can cause significant concern among consumers [9].

Dependencyon systems without proper knowledge of execution can pose a great threat to this sector. For instance, minor security breaches may often lead toimmense financial losses. For better performance and output optimization, adequate training, and counseling for information literacy are grievous necessities for the banking system which requires the knowledge of Service Supply Chain Management (SCM). Service SCM system assists service enterprises by optimizing the core businesses, minimizing expenses, improving service quality, and so on [10]. With the help of modern technology, service SCM has successfully strengthened the banking business throughout the world. Countries around the globe are taking technology-based fiscal measures and adoptingan extensive monetary policy to evade possible economic contraction.

The banking service sector interacts with several other sectors for the growth of the economy. Banks and other NBFIs hold a major share in Bangladesh's economy. Including a total of 50 national banks with 9 international banks and 34 NBFI, the sector shows exquisite growth promises in the country's economy. In the era of IT, banking in Bangladesh is not merely confined to the banks; along with the development of information management and communication technology, financial institutions are executing transactions via deploying agents as well as smartphone-based applications for their customers. In recent years, as a major part of the service supply chain (SSC), IT-based banking has spread all around the country by means of various products like online banking, mobile banking, agent-based banking, etc. Now, people have embraced internet banking more than ever and the coverage of these services is spreading more and more, considering their extensive demands. The authorities of the financial organizations are also encouraging fund transfer through internet banking to uplift those various IT-based banking servicesand improve the quality of those services [11]. Extended transaction limit, ceiling per transaction, and transactions per day are some of the steps taken by the banks to promote IT-based banking services.

However, it has been observed that, even though the rest of the world is well aware of the safety and security of IT-based banking,the banking sector,especially in Bangladesh, is still struggling with it. Although technology being a propelling factor of the economy, there exist threats and failures to safeguard the business from various existing loopholes [12].Clementina and Isu [13] evaluate the insecure situation, bank fraud, and their impact on bank performance fromthe perspective of the commercial banks of Nigeria. The study used a multiple regression analysis to determine if there is any significant relationship between the indicators of bank insecurity and fraud. Ula et al. [14] explores the relationship between information assets and potential threats tothe banking system. The study also examines and compares the elements from the commonly used information security governance frameworks, standards, and best

practices. Edge et al. [15] tried to help the banks and other financial institutions to identify how attackers compromise accounts and develop methods to protect them. They used an 'attack trees and protection trees' method to do this.

Various MCDM techniques have been used in the area of failure and risk analysis in recent time. For example, Bathrinath et al. [16] analyzed the risks in the textile industry using an Analytic Hierarchy Process (AHP)- Technique for Order of Preference by Similarity to Ideal Solution (TOPSIS) hybrid method. Şenel et al. [17] analyzed the risks in the maritime industries of Turkey using FMEA based intuitionistic Fuzzy TOPSIS Approach. Pamučar et al. [18] used a multi-criteria Full Consistency Method (FUCOM)- Multi-Attributive Ideal-Real Comparative Analysis (MAIRCA) model for the evaluation of level crossings in the Republic of Serbia. Stević and Brković [19] utilized a hybrid FUCOM- Measurement of alternatives and ranking according to compromise solution (MARCOS) model for evaluation of human resources in a transport company. Jokić et al. [20] used a Level Based Weight Assessment (LBWA)-Fuzzy Multi-Attributive Border Approximation area Comparison (MABAC) method for the selection of appropriate firing positions for the mortars used by the military artillery unit. Liu et al. [21] used an integrated Stepwise Weight Assessment Ratio Analysis (SWARA)- MABAC method to assess occupational health and safety risk. Hou et al. [22] analyzed the safety risks in the metro construction under epistemic uncertainty, using credal networks and the Evaluation Based on Distance from Average Solution (EDAS) method. Bakhat and Rajaa [23] analyzed the risks in a wind turbine operation in Morocco using a Gray AHP-MABAC approach. Xu [24] performed a performance evaluation in the investment environment of blockchain industry using a Fuzzy Combinative Distance based Assesment (CODAS) method. However, there has not been any significant research using any MCDM technique on the identification and analysis of the factors contributing to the IT failures in the in financial institution so far, which presents a clear research gap.

Hence, this research, at first, intends toidentify the factors that contribute to the failure of the banking IT systems from expert feedbacks and previous relevant literatures. After that, it proposes a rough-TOPSIS (Technique for Order of Preference by Similarity to Ideal Solution) based flexible Failure Mode and Effect Analysis (FMEA) approach to evaluate the identified factors.

This research was motivated by therecent banking security failure incidents that took place in Bangladesh. The 2016 cyber heist, or the recent automated teller machine (ATM) theft of 2019, corroborates that the financial sector of Bangladesh is on the verge of security abuse. In June 2019, nine ATM booths of a bank have encountered a scam that incurred a loss of Tk 1.4 million [25]. Experts are still not sure whether the hacker syndicate exploited the server issue or if it is the fault of the bank's ATM software, which in some way proves the lack of literacy in the bank management.

Moreover, unscrupulous officials and business personnel collaborated to exploit the Letter of Credit (LC) system with forged documents in the name of bogus companies. The LC scam of Tk 36.48 billion between 2010 and 2012 is one of such notable incidents [26]. In 2016, another setback hit the financial sector of Bangladesh, which eventually resulted in the transfer of USD $101 million to two countries by infecting the system with a malware. Although $38 million has been fully recovered from two different countries later, the remaining $63 million is yet to be recovered [25]. In the case of local transactions,the Society for Worldwide Interbank Financial Telecommunication (SWIFT) network is connected with the Real Time Gross Settlement (RTGS) system. Hackers exploited the vulnerability of the SWIFT-RTGS connection illegally for the cyber theft incident. That incident is still well known as the biggest-ever cyber heist in Asia. Unlike traditional hacking of the account holders' login credentials, this attack targeted the Bank's credentials by infecting their systems with malware. Incidents like

this have shaken up the trust of the customers in the IT-based banking system security as a whole.

Automation and upgradation of the management information systems were some of the critical suggestions from the security expertsto resolve these alarming security issues of Banks and NBFIs. Addressing technical issues and business confidentiality is not enough and a continuous effort to keep an updated security system is essential. Improving preventive measures can minimize the collateral damages, instead of regular troubleshooting.

Several studies have suggested that in many case customers demonstrates aversion in accepting IT-based business activities, largely because of security concerns and other related trust issues [27]. Because of the security issues in the financial industries of Bangladesh, there is an escalated demand for technical security and information management system [28]. In this regard, a managerial survey and sector-wise analysis under the concept of SSC are required to be performed [29]. However, the selection of an appropriate mathematical model for the evaluation of SSC properties is challenging due to the qualitative nature of these failures [30]. Meanwhile, monetary, reputational, as well as information loss [31], can occur if any of the failure factors remains unchecked. Therefore, experts have suggested that identifying and ranking the failure factors can help them to prioritize the issues that need to be addressed to prevent these security failures from happening again in the future [32].

Previous literature, however, hardly provides any concrete insights that recognize and rank various failure factors of this SSCespecially in terms of IT systems of the banking sectorof Bangladesh. Despite the groundbreaking growth of this industry, there have been many voids in previous studies in the establishment of an effective model to address these problems in the context of Bangladesh. Therefore, this work aims to cover the research gap that targets the financial sector when it is vulnerable to IT-related security abuse. The main purpose of this research is to recognize as well as analyze the failure issues of the IT systems in the field of the SSC in the Banking sector of Bangladesh.

In this study, a rough TOPSIS based FMEA approach has been used for effective identification and prioritization of the most significant failures. FMEA and TOPSIS variants have been used together before in several recent studies involving failure and risk analysis. For example, Vahdani et al. [33] utilized this approach to assess the failure causes of the steel production process; and Selim et al. [34] developed a dynamic maintenance planning framework for an international food company. Recently, Başhan et al. [35] used these for maritime risk evaluation and ship navigation safety.

A rough TOPSIS method has been used here, which combines rough set theory with the traditional TOPSIS method [36]. The Rough Set theory addresses the uncertainty of human judgments, where performance rating and weights cannot be assigned accurately [37]. Hence, in this study, the framework integrates the strength of rough set theory to tackle vagueness and the merit of the TOPSIS assessment structure. It is used in most cases where the study involves dealing with imprecise or incomplete information [38]. For instance, this method has been used successfully for supplier selection [39], career path selection for students [40], parametric analysis for the machining process [41] and so on. The reason rough TOPSIS is often preferred in much recent research is that it not only improves the reliability of the TOPSIS calculation program but also expresses more potential information considering the uncertainties [36,42],. The proposed rough TOPSIS based on flexible FMEA evaluates the failure modes except for prior information and made the execution of the FMEA process very effective [43].

To implement the proposed rough-TOPSIS framework, a case study on several state-owned as well as private commercial banks and NBFIs in Bangladesh has been carried out. This study collected information from several banks and software firms of Bangladesh intending to formulate a framework to prioritize the predetermined failure issues. Experts of various

disciplines have shared their experiences focusing on the main essence of this study. The goal is to gather and share valuable experiences and knowledge for the development of the Banking IT systems that can assist in minimizing the security risks in this sector. The rest of the paper is organized as follows. Section 2 discusses the related materials and methods. Section 3 presnts the results and discusses them. Section 4 concludes this paper.

## 2 Materials and methods

### 2.1 Conceptualization of IT failure factors

This research conceptualizes the factors contributing to the failure of the IT system in the banking industry. Based on extant literature and expert inputs, several factors were identified. The role of each factor is discussed below.

**Database hack.** For any service, the supply chain database comprises comprehensive links, which can be used to analyze organization based risks [44]. Renowned companies around the world, heavily rely on their centralized database servers. For IBM, to serve the business points simultaneously with negligible slack, their database is considered as part and parcel of the company [45]. Any hindrance to the process can compromise the whole information system. Loss of login credentials, unauthorized changes in settings, and other vital information may pose threats to IT-based businesses. Hereby, routine backup of the database can mitigate the impact of database hack [46].

**Server failure.** The server is one of the vital parts as potential hackers sneak into it or infect to serve their heinous purposes. In 2002, an international financial services company of the United States lost its 10 billion files that eventually affected more than 1300 companies' servers [47]. One failure of such kind may often lead to many other failures of various types like the ripple effect. Prompt preventive measures, timely backups, and proper recovery maneuvers can minimize the impact of these losses [48]. Hence, failure of the server may result in loss of information and thus pose a vital threat to the business.

**Virus effect.** Now-a-days organizations related to IT as well as financial sectors are susceptible to virus attacks. Malware can be divided into two broad categories: network-based and non-network based. In fact, the 2016 cyber heist was initiated through malware. Cloud-based supply chain management systems are getting more popular day by day, even though there are still chances of data damage through viruses [49]. In order to minimize the impact of viruses, a wide range of steps like surveillance on IT systems, mechanisms similar to proxy server code repositories, regular scanning of the system, etc. should be implemented [50].

**Cipher to plaintext malfunction.** Improper interpretation of cipher-text may lead to wrong decryption of a message. Though cryptography protects sensitive data by encryption [51], if deciphering is not executed according to the key, plaintext remains undiscovered. This way, the main purpose of encryption hardly serves.

**Character misspelled.** The misspelling character of a message can often create great confusion. For example, if a bank's IT official misspells a decimal, it may result in enormous financial loss. Hence, people working in the Banking IT system have to be extremely careful about it.

**Wrong message transcription.** Unlike other significant failures, message transcription may seem trivial, yet has a significant impact on proper communication among various business and banking entities. Especially, in the IT-based financial industries, message transcription is highly sophisticated. Miscommunication, thereby, can cause huge financial losses.

**Peripheral error.** It involves the unintentional errors that occur due to erroneous use of input and output devices, which can create failure of the IT system of the Banking or Financial industry. It has been observed that unskilled and inept users in the bank (bank officials and staff), often being unaware of the proper usage technique of IT system devices, use the input

and output devices of the system in an improper way, which is usually a major reason for peripheral errors.

**Broadcast data missing (up/down) link failure.** Larger the network, the higher the chances of link failure. Accidental disconnections and electromagnetic interferences negatively impact the network's reliability. Up/down link failure, thereby, impairs overall network performance [52].

**Cyber attack.** IT-based businesses like banking industries, IT companies, and other financial institutions put utmost importance on cybersecurity. Once the malware infects the system, it can be used to trigger secretly and anonymously for unauthorized action [53]. Cybercriminals with ill intentions are a grave risk to SSC security all over the world.

**Third party intervention.** An analysis of information system risk identifies deliberate external database attacks as the vital risks [54]. Human failure is the prime reason for third party intervention, which can be categorized as security abuses. IT companies or software organizations are dependent on vendors, and some other third parties to some extent. A security breach can occur from the end of the third parties if the financial institutions are not very careful.

**Network interruption.** Although a survey on the US-based bank implied return on asset (ROA) and network system variables to be mutually independent [5], banking activities nowadays are predominantly dependent on IT, including all the online transactions largely. Hence, even a slight failure in the IT/IS network can trigger doubts about a large number of transactions [55]. Table 1 summarizes the literature reviewed for identifying the failure factors considered in this study.

## 2.2 Research steps

This research work focuses on identifying the failure factors and prioritizing them in the context of the banking industry of Bangladesh. Fig 1 illustrates the steps followed in this research.

The proposed research consists of five steps, as mentioned below.

Step 1:Identification of failure factors

The objective of this step is to generate a comprehensive list of failure factors of the Banking IT system, based on the events that might hurt the banking industry. In this step, 3 relevant SSC failure factors are identified from experts' input and 8 factors have been identified from the previous relevant studies. Thus, a total of 11 failure factors of the IT system in the banking industry have been identified from surveying the experts and analyzing the literature in the corresponding field (see Table 1). After that, the crisp weights of failure factors have been determined.

**Table 1. Failure factors of IT System in the Banking Industry.**

| No. | Failure factors | Source |
|-----|-----------------|--------|
| FM1 | Database Hack | [44–46] |
| FM2 | Server failure | [47,48] |
| FM3 | Virus Effect | [49,50] |
| FM4 | Cipher to Plain Text Malfunction | [51] |
| FM5 | Character Misspelled | Proposed in this research |
| FM6 | Wrong Message Transcription | Proposed in this research |
| FM7 | Peripheral Error | Proposed in this research |
| FM8 | Broadcast Data Error (Up/Down) link Failure | [52,56] |
| FM9 | Cyber Attack | [53] |
| FM10 | Third Party Intervention | [54] |
| FM11 | Network Interruption | [5,55] |

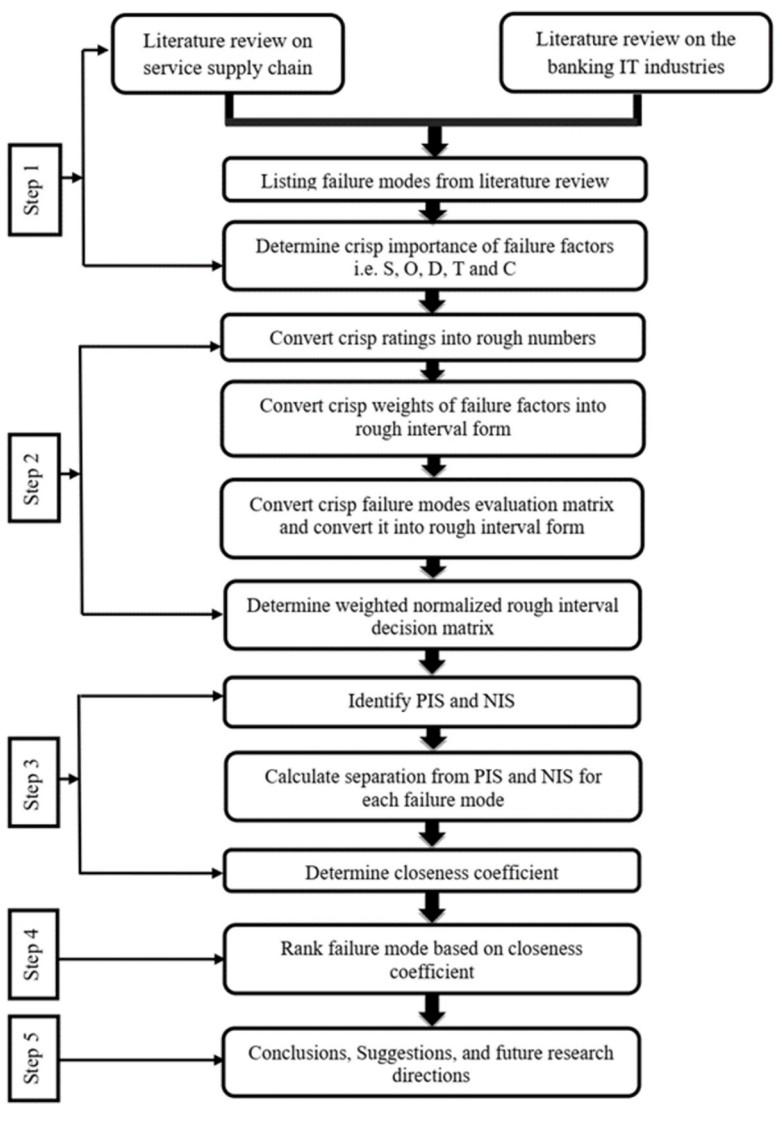

**Fig 1. Research steps.**

Step 2: Conversion of the rough interval by using the rough TOPSIS method

In this step, the crisp importance of failure factors is converted into a rough interval form. Then rough TOPSIS based flexible FMEA approach has been implemented by converting constructed group decision matrix into the rough interval. Afterward, a weighted rough interval decision matrix has been determined. The decision matrix uses the weighted normalization process to formulate a corresponding rough number.

Step 3: Determination of closeness coefficient

In this stage, the identification of positive ideal solution (PIS) and negative ideal solution (NIS) is executed. Then separation from PIS and NIS for failure modes is calculated. After that, a closeness coefficient is defined. It is determined with respect to the criteria like Severity (S), Occurrence (O), Detection Difficulty (D), Time (T), and Cost (C).

Step 4: Ranking of failure modes

Failure factors are ranked according to their importance. The closeness coefficient (CC) is the basis of this ranking.The managers will address the issues according to the risk ranking.

Step 5: Conclusions, managerial implications, and recommendations

Implementation of this model is mainly applicable when banking managers use it for solving problems that have been generated from IT failures. The future scope and limitations of this work have also been discussed in this section.

## 2.3 Rough set theory

Rough set theory has applications in many areas of research. One of the most important applications of rough set theory is that it's incorporation eliminatesthe impact of vagueness in decision making [37]. For example, in the area of decision analysis, the decision-makers are required to evaluate the criteria for a particular problem and provide feedback on them using some particular scaled values. Since it is not always possible to make sure that all the decision-makers are experts in all fields, an inexperienced decision-maker can decide on a particular area and the judgment made by that expert might contain uncertainty. In order to find and eliminate these uncertainties, rough set theory plays an important role [43]. Basic equations of rough set theory and rough number are presented in the Appendix.

## 2.4 Background of rough TOPSIS method based on flexible FMEA

To prioritize the failure modes of IT systems in the banking sector, a flexible FMEA approach has been presented in this research. In conventional FMEA,possible indirect relationships between the factors are not considered and the failure factors are weighted equally; whereas, if risk analysis cases differ, their weights should also vary [42].The scale used by conventional FMEA depends on absolute values. Experts often face difficulties due to the lack of historical data [33]. Flexible FMEA has been used in this research,since it can overcome the above-mentioned issues of conventional FMEA. Flexible FMEA is a relatively new approach where the risks that are identified in multiple FMEAs or studies are combined to provide a complete picture or information. This technology includes the extraction of risk information envisioned at the beginning of the study as the result of unpredicted incidents in a process. To deplete process diversity, flexible FMEA has been proven to be useful in recognizing opportunities.

In this research, the rough TOPSIS method based on flexible FMEA follows the following steps.

**Step 1:** Formation of the expert panel

Experts with various range of experiences were selected from diversified but related fields, which includes professionals from the bank and other NBFI, IT specialists and engineers.

**Step 2**: Determining rough interval weight of failure factors S, O, D, T, and C

(1) After determining the failure modes, the experts are required to choose the crisp importance of each criterion (S, O, D, T, and C) using a scale of scores from 1 to 10. A score of 1 indicatesthat the criterion is of the least significance, while a score of 10 demonstrates extremelyrelevance. Therefore, a crisp evaluation value can be achieved for failure factor's weight.

$$w_j = [w_j^1, w_j^2, \ldots, w_j^k, \ldots, w_j^l]; j = \text{S, O, D, T, and C} \qquad 1$$

Where $w_j^k$ represents the $k^{\text{th}}$ expert assessment on the significance of the criterion of $j$. $l$ represents the number of experts in the decision matrix.

(2) The rough number form is then derived from crisp importance with the formula in Appendix A in S1 File. The rough interval form of $w_j^k$ can berepresented as

$$RN(w_j^k) = [w_j^{kL}, w_j^{kU}] \qquad 2$$

Where $L$ and $U$ represent the lower limit and upper limit of rough number $RN(w_j^k)$

The following equations are used to determine the rough weight of criterion$j$ $\overline{RN(w_j)}$.

$$w_j^L = \frac{(w_j^{1L} + w_j^{2L} + \ldots + w_j^{lL})}{l} \qquad\qquad 3$$

$$w_j^U = \frac{(w_j^{1U} + w_j^{2U} + \ldots + w_j^{lU})}{l} \qquad\qquad 4$$

$j$ = S, O, D, T, and C; $w_j^L$ and $w_j^U$ are lower limit and upper limit of rough weight $\overline{RN(w_j)}$ respectively.

**Step 3**: Rough TOPSIS framework based on flexible FMEA approach

(1) Construction of crisp failure modes evaluation matrix: At first, it is assumed that there are $m$ failure modes $FM_i$ ($i = 1,2, \ldots, m$) which is to be evaluated against assigned criteria $C_j$ ($j$ = S, O, D, T and C). Failure mode ratings with respect to criteria are then evaluated from multidisciplinary experts' input on conventional scores, in this case from 1 to 10; where 10 indicates the most important and 1 indicates the least important. Assuming that$l$ experts of an FMEA team are making decisions, the failure modes ranking in FMEA can be expressed in the form of evaluation matrix $D$, which can be written as follows:

$$
D = \begin{array}{c}
\phantom{FM_1}\quad\ \ \, \text{S}\quad\ \ \text{O}\quad\ \ \text{D}\quad\ \ \text{T}\quad\ \ \text{C} \\
\begin{array}{c} FM_1 \\ FM_2 \\ \vdots \\ FM_m \end{array}
\begin{bmatrix}
X_{1S}^k & X_{1O}^k & X_{1D}^k & X_{1T}^k & X_{1C}^k \\
X_{2S}^k & X_{2O}^k & X_{2D}^k & X_{2T}^k & X_{2C}^k \\
\vdots & \vdots & \vdots & \vdots & \vdots \\
X_{mS}^k & X_{mO}^k & X_{mD}^k & X_{mT}^k & X_{mC}^k
\end{bmatrix}
\end{array} \qquad\qquad 5
$$

Where $k = 1,2,\ldots,l$and $X_{ij}^k$ ($i = 1,2,\ldots,m$) is the rating of the $k$th expert for the $i$th failure mode with respect to the criterion $j$.

(2) Obtaining Rough group decision matrix: To obtain rough group decision matrix$R$, the crisp element $X_{ij}^k$ in the group decision matrix $D$ can be converted into rough number form. Rough number form $RN(X_{ij}^k)$ of $X_{ij}^k$ can be executed with the help of equations listed in appendix A in S1 File.

$$RN(X_{ij}^k) = [X_{ij}^{kL};\ X_{ij}^{kU}] \qquad\qquad 6$$

Where $X_{ij}^{kL}$ and $X_{ij}^{kU}$ represent the lower limit and upper limit of rough number $RN(X_{ij}^k)$ respectively. Hence, a rough number form$m$can be achieved as follows.

$$RN\ (X_{ij}) = \{[X_{ij}^{1L};\ X_{ij}^{1U}], [X_{ij}^{2L};\ X_{ij}^{2U}], \cdots, [X_{ij}^{lL};\ X_{ij}^{lU}]\} \qquad\qquad 7$$

Using rough computation principles, the average rough interval can be acquired.

$$\overline{RN(X_{ij})} = [X_{ij}^L; \ X_{ij}^U]$$

$$X_{ij}^L = \frac{(X_{ij}^{1L} + X_{ij}^{2L} + \ldots + X_{ij}^{lL})}{l}$$

$$X_{ij}^U = \frac{(X_{ij}^{1U} + X_{ij}^{2U} + \ldots + X_{ij}^{lU})}{l}$$

Thus, the rough group decision matrix *R* can be obtained as follows:

$$R = \begin{bmatrix} [1,1] & [X_{12}^L, X_{12}^U] & \cdots & [X_{1m}^L, X_{1m}^U] \\ [X_{21}^L, X_{21}^U] & [1,1] & \cdots & [X_{2m}^L, X_{2m}^U] \\ \vdots & \vdots & \ddots & \vdots \\ [X_{m1}^L, X_{m1}^U] & [X_{m2}^L, X_{m2}^U] & \cdots & [1,1] \end{bmatrix}$$

(3) Determination of weighted normalized decision matrix in the form of rough number: Following equations show how the normalization method is used to transform different criteria scales into a comparable scale:

$$X_{ij}'^L = \frac{X_{ij}^L}{max_{i=1}^m\{\max[X_{ij}^L, X_{ij}^U]\}}$$

$$X_{ij}'^U = \frac{X_{ij}^U}{max_{i=1}^m\{\max[X_{ij}^L, X_{ij}^U]\}}$$

The method mentioned earlier regarding the normalization is designed to preserve the property of normalized interval numbers' ranging between [0, 1]. Afterward, the weighted normalized rough matrix can be calculated as follows:

$$V_{ij}^L = W_j^L \times X_{ij}'^L, \ i = 1, 2, \ldots, m; j = S, \ O, \ D, \ T, \ and \ C$$

$$V_{ij}^U = W_j^U \times X_{ij}'^U, \ i = 1, 2, \ldots, m; j = S, \ O, \ D, \ T, \ and \ C$$

(4) Now, Positive Ideal Solution (PIS) and Negative Ideal Solution (NIS) can be obtained as follows

$$V^+(j) = \{max_{i=1}^m(V_{ij}^U), \ if \ j \in B; \ min_{i=1}^m(V_{ij}^L), \ if \ j \in C\}$$

$$V^-(j) = \{min_{i=1}^m(V_{ij}^L), \ if \ j \in B; \ max_{i=1}^m(V_{ij}^U), \ if \ j \in C\}$$

Where $V^+(j)$ and $V^-(j)$ are PIS and NIS values with respect to criterion *j*. B and C are associated with the benefit criterion and cost criterion, respectively.

(5) Using the *n*-dimensional Euclidean distance equation, the separation of individual failure mode from the PIS can be calculated as follows.

$$d_i^+ = \sqrt{\left\{\sum_{j \in B}(V_{ij}^L - V^+(j))^2 + \sum_{j \in C}(V_{ij}^U - V^+(j))^2\right\}}$$

Likewise, the separation from the NIS can be calculated as

$$d_i^- = \sqrt{\left\{ \sum_{j \in B} \left( V_{ij}^U - V^-(j) \right)^2 + \sum_{j \in C} \left( V_{ij}^L - V^-(j) \right)^2 \right\}} \qquad 19$$

Once the $d_i^+$ and $d_i^-$ of individual failure modes $FM_i$ have been calculated, a closeness coefficient is defined for determining the ranking order of identified failure modes. The closeness coefficient $CC_i$ of the failure modes $FM_i$ with respect to selected criterion (S, O, D, T, and C) is defined as

$$CC_i = \frac{d_i^-}{d_i^- + d_i^+} \qquad 20$$

As $CC_i$ approaches to 1, failure modes $FM_i$ gets closer to the $d_i^+$ and farther from $d_i^-$. The smaller the $CC_i$ is, the severe risk of failure mode becomes. After that,the risk priority of identified failure modes can be determined in consistence with the closeness coefficient.

## 2.5 Data

Data were collected from the banking industry of Bangladesh. Experts in the IT and software industry with years of experience in the banking industry have been selected as usual candidates for the survey. Existing literature, as well as 32 experts of versatile fields, have identified major failure factors. Eventually, they participated in ranking them as per the proposed methodology of the previous section. Experts spontaneously participated in the non-disclosure survey.

In this study, two steps have been followed for collecting the necessary data and information. In Step 1, based on the literature review and managers' opinion, factors contributing to failure modes of IT systems in the banking industrywere identified, and in Step 2, the analysis of the identified failure factors with the help of experts' input was performed.

Before the data collection phase, a specialist panel from the pertinent industrieswho have multidisciplinary professional experience was formed. Afterward, the required data have been collected from the experts. The survey was initially conducted, using a questionnaire, both online and offline as per the professionals' convenience. Their feedback was documented to develop and clarify the questionnaire. All the experts who took part in this study are either bank officials/personnel or IT professional, who hold at least a Masters degree in their relevant area of expertise. The majority of the participating IT professionals have degree in computer engineering. However, the experts were not comfortable to share and publish their exact academic background. Therefore, this information is not included in this study. Abrief summary of the experts based on their experience is listed in Table 2.

The two-step data collection process is explained as below:

***Step 1: Recognizing factors responsible for IT system failure in banks and NBFI***

**Table 2. The domain of experts.**

| Domain of Work | Years of Experience | | | |
|---|---|---|---|---|
| | Up to 5 | 5–10 | More than 10 | Total |
| IT specialist of Financial Institutions | 7 | 7 | 3 | 17 |
| Financial Institutions | 2 | 3 | 1 | 6 |
| IT professionals | 2 | 4 | 3 | 9 |
| Total | 11 | 14 | 7 | 32 |

At first, from the previous history and reports from the related organizations, factors responsible for IT system failure in banks and NBFI have been identified. During the primary phase, the experts' panel was requested to make necessary modifications or inclusion of any risk relevant to the failure of the IT system in the financial sector of Bangladesh. Subsequently, the responses were gathered from the experts in order to finalize the list. This way, eleven possible factors were identified (Table 1) that are responsible for the IT system failure of the financial sector in Bangladesh

***Step 2*: *Analysis of identified failure modes with the help of experts' opinions for ranking***

The identified failure factors were ranked by implementing rough set theory and TOPSIS. For the analysis, a meeting with the expert panel was arranged at the very beginning to generate a basic idea. With the help of their feedback, crisp importance criteria and failure modes were listed in a table for the survey. After that,using the experts' input, the ranking of failure modeswas performed.

Once the major potential failure modes are determined, subjective assessments of 32 experts in crisp variables are utilized to obtain the importance of failure factors (S, O, D, T, and C), which are determined according to the basic formula of rough set theory and rough number. The experts' perspectives on crisp ratings about failure modes regarding each failure factor are then determined.

# 3 Results and discussion

## 3.1 Results

### 3.1.1 Rough interval and normalized weight.
At first, the evaluating criteria are rated based on expert opinions. Ratings were scaled from 1 to 10.Score 1 means the least importance, while a score of 10 indicatesthe highest importance. Failure modes are then rated based on different criteria and expert opinions.

Using the formula of the lower limit and upper limit listed in the Table A1 of Appendix A in S1 File, a rough number form of the crisp importance rating is calculated. A sample calculation, for better comprehension, is presented below. Let the crisp ratings for failure factor 'Severity' according to 4 experts are [2, 4, 7, 7].

| Lower Limit | Upper Limit | Rough Interval |
|---|---|---|
| $\underline{Lim}(2) = 2$ | $\overline{Lim}(2) = \frac{1}{4}(2 + 4 + 7 + 7)$ | [2,5] |
| $\underline{Lim}(4) = \frac{1}{2}(2 + 4)$ | $\overline{Lim}(4) = \frac{1}{3}(4 + 7 + 7)$ | [3,6] |
| $\underline{Lim}(7) = \frac{1}{4}(2 + 4 + 7 + 7)$ | $\overline{Lim}(7) = 7$ | [5,7] |
| | Average Rough Interval | [3.75,6.25] |

As per equations of appendix A in S1 File and Eqs 3 and 4, the average rough intervals are determined. Likewise, for different failure factors, rough number forms, and average rough intervals can also be acquired, as shown in Table 3.

### 3.1.2 Rough TOPSIS-based flexible FMEA ranking of the failure modes.
The failure modes' crisp decision matrix is converted as a rough group decision-making matrix; hence the failure modes' rough interval evaluation is determined as shown in Table 4.

Afterward, the rough form of a weighted normalized decision matrix is obtained. The evaluation matrix of rough failure modes is normalized in Table 5, and the weighted normalized

**Table 3. The rough interval and normalized weights for S, O, D, T, and C.**

| Risk Factor | Rough interval | Normalized rough weight |
|---|---|---|
| | [Low High] | [Low High] |
| Severity | [6.422 8.949] | [0.680 0.947] |
| Occurrence | [6.619 9.446] | [0.701 1.000] |
| Detection Difficulty | [6.356 8.880] | [0.673 0.940] |
| Time | [6.080 8.418] | [0.644 0.891] |
| Cost | [4.985 7.626] | [0.528 0.807] |

**Table 4. The evaluation matrix of rough failure modes.**

| No. | S | O | D | T | C |
|---|---|---|---|---|---|
| FM1 | [7.989 9.178] | [5.566 8.946] | [6.200 9.467] | [6.726 9.458] | [5.320 8.671] |
| FM2 | [7.225 9.480] | [5.866 8.719] | [5.486 8.911] | [6.246 9.326] | [5.790 9.169] |
| FM3 | [5.728 7.779] | [4.693 7.613] | [5.389 8.077] | [5.797 7.896] | [5.110 8.073] |
| FM4 | [4.521 7.406] | [4.556 7.230] | [4.719 7.256] | [4.336 7.127] | [3.744 6.600] |
| FM5 | [4.415 7.629] | [4.479 7.207] | [4.302 6.496] | [3.929 6.815] | [3.489 6.386] |
| FM6 | [5.451 7.965] | [4.121 7.190] | [4.335 6.876] | [4.822 7.605] | [3.917 6.735] |
| FM7 | [4.491 6.868] | [4.251 6.986] | [3.639 6.886] | [3.914 6.377] | [3.493 6.353] |
| FM8 | [5.647 8.331] | [5.027 8.366] | [4.290 8.302] | [5.001 7.653] | [4.129 6.791] |
| FM9 | [8.334 9.598] | [6.249 9.207] | [7.596 9.496] | [6.374 8.870] | [6.402 9.079] |
| FM10 | [5.682 8.209] | [4.794 7.591] | [3.999 8.137] | [5.331 7.724] | [4.676 6.939] |
| FM11 | [6.628 8.709] | [6.121 8.463] | [5.507 8.354] | [5.677 8.221] | [4.444 8.281] |

**Table 5. The rough weighted normalized matrix of failure modes.**

| Weighted Matrix | Severity | | Occurrence | | Detection | | Time | | Cost | |
|---|---|---|---|---|---|---|---|---|---|---|
| | Low | High | Low | High | Low | High | Low | High | Low | High |
| FM1 | 0.566 | 0.906 | 0.424 | 0.972 | 0.439 | 0.937 | 0.458 | 0.891 | 0.306 | 0.763 |
| FM2 | 0.512 | 0.936 | 0.446 | 0.947 | 0.389 | 0.882 | 0.425 | 0.879 | 0.333 | 0.807 |
| FM3 | 0.406 | 0.768 | 0.357 | 0.827 | 0.382 | 0.800 | 0.395 | 0.744 | 0.294 | 0.711 |
| FM4 | 0.320 | 0.731 | 0.347 | 0.785 | 0.334 | 0.718 | 0.295 | 0.672 | 0.215 | 0.581 |
| FM5 | 0.313 | 0.753 | 0.341 | 0.783 | 0.305 | 0.643 | 0.267 | 0.642 | 0.201 | 0.562 |
| FM6 | 0.386 | 0.786 | 0.314 | 0.781 | 0.307 | 0.681 | 0.328 | 0.717 | 0.225 | 0.593 |
| FM7 | 0.318 | 0.678 | 0.324 | 0.759 | 0.258 | 0.682 | 0.266 | 0.601 | 0.201 | 0.559 |
| FM8 | 0.400 | 0.822 | 0.383 | 0.909 | 0.304 | 0.822 | 0.340 | 0.721 | 0.238 | 0.598 |
| FM9 | 0.590 | 0.947 | 0.476 | 1.000 | 0.538 | 0.940 | 0.434 | 0.836 | 0.368 | 0.799 |
| FM10 | 0.402 | 0.810 | 0.365 | 0.824 | 0.283 | 0.806 | 0.363 | 0.728 | 0.269 | 0.611 |
| FM11 | 0.469 | 0.860 | 0.466 | 0.919 | 0.390 | 0.827 | 0.386 | 0.775 | 0.256 | 0.729 |

rough matrix is then determined. In Table 5, eleven failure modes' rough weighted normalized matrix is depicted.

PIS and NIS are identified using Eqs 16 and 17. The failure factors S, O, D, T, and C are all related to cost criterion according to the framework that the flexible FMEA approach proposed. In Table 6, the PIS and NIS are demonstrated.

Then the segregation of each failure mode from the PIS and NIS was calculated.

**Table 6. The PIS and NIS of the rough weighted normalized matrix.**

|  | Severity | Occurrence | Detection | Time | Cost |
|---|---|---|---|---|---|
| PIS | 0.313 | 0.314 | 0.258 | 0.643 | 0.266 |
| NIS | 0.936 | 1.000 | 0.940 | 0.458 | 0.891 |

According to the assumptions of the risk priority number method, based on the flexible FMEA, the weights of the five failure factors are considered to be of equal crisp value. In the process of determining the weights of failure factors, the crisp value-form of weights lacks the subjectivity and ambiguity inherent in it. Table 7 highlights the closeness coefficient and rank of the failure modes.

## 3.2 Model comparison

Fig 2 presents the graphical representation of a comparison of weights of severity, occurrence, detection, time, and cost obtained by the rough method to the conventional crisp method. It is noteworthy to mention that the order of ranking of weights of all the factors by the rough method is almost the same as the rank order of the crisp method. Occurrence > Detection > Severity > Time > Cost is the rank order obtained by rough method while the sequence of rank by crisp method is Occurrence > Severity > Detection > Time > Cost.

However, the rough method is effective to represent the uncertainties as it fits the values of decision-makers in the form of upper and lower limits. According to Fig 2, the spread of judgment by the experts is represented in the form of the bar for the rough method process as opposed to a line by the crisp method. The less the length of the bar indicates, the less the uncertainties of decisions by the decision-makers. The more the length of the spread represents, the lower the accuracy of the decisions. When it comes to the weights by the traditional crisp method or other MCDM methods like Analytic Hierarchy Process (AHP), Best-Worst Method (BWM), they are represented by a single crisp value or in the form of lines shown in Fig 2, although multiple decision-makers were involved in this decision making. All these methods consider only the mean decision value by the experts and the vagueness and uncertainties of the judgment values cannot be represented properly by these methods.

The rank of the failure modes found by the rough TOPSIS method is also compared with the rank of failure modes by the crisp TOPSIS method and presented in Fig 3. It can be seen from the graph that cyber attack is the most critical failure factor based on both methods. The ranking of database hack, server failure, and network interruption are most similar for both

**Table 7. $d_i^+$, $d_i^-$, closeness coefficient and rank of each failure mode.**

| No. | Failure mode | $d_i^+$ | $d_i^-$ | $CC_i$ | Rank |
|---|---|---|---|---|---|
| FM1 | Database Hack | 1.116 | 0.848 | 0.432 | 2 |
| FM2 | Server failure | 1.086 | 0.889 | 0.450 | 3 |
| FM3 | Virus Effect | 0.874 | 1.003 | 0.534 | 6 |
| FM4 | Cipher to Plain Text Malfunction | 0.781 | 1.083 | 0.581 | 9 |
| FM5 | Character Misspelled | 0.750 | 1.107 | 0.596 | 10 |
| FM6 | Wrong Message Transcription | 0.788 | 1.083 | 0.579 | 8 |
| FM7 | Peripheral Error | 0.715 | 1.142 | 0.615 | 11 |
| FM8 | Broadcast Data Error (Up/Down) link Failure | 0.965 | 1.036 | 0.518 | 5 |
| FM9 | Cyber Attack | 1.157 | 0.746 | 0.392 | 1 |
| FM10 | Third Party Intervention | 0.899 | 1.058 | 0.541 | 7 |
| FM11 | Network Interruption | 0.995 | 0.897 | 0.474 | 4 |

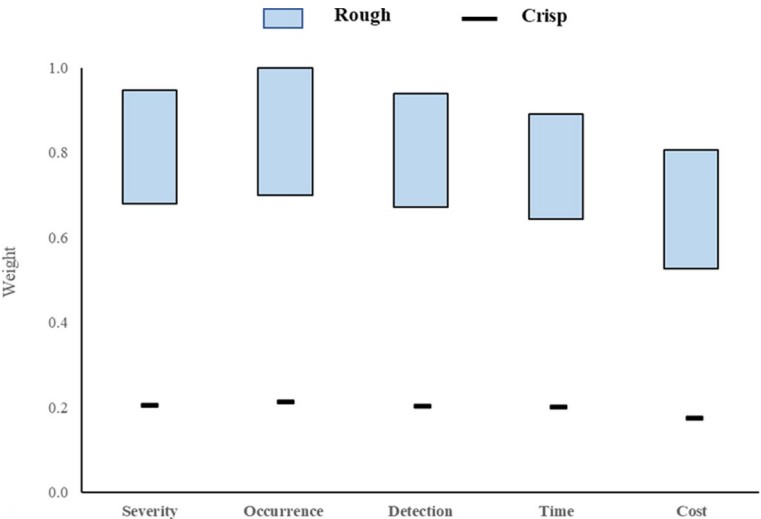

**Fig 2. Comparison of weights using the rough and crisp value.**

methods. There exists slight ranking variation for the cipher to plain text malfunction and broadcast data error factors. However, peripheral error and character misspelled factors show the most significant difference. It ranked fifth based on the crisp TOPSIS method while eleventh based on the rough TOPSIS method. Similarly, character misspelled ranked sixth and tenth based on the crisp TOPSIS method and rough TOPSIS method, respectively. According to the crisp TOPSIS method, wrong message transcription is the least critical factor whereas the rough TOPSIS method indicates the peripheral error. The results provided by the rough TOPSIS method are more reliable and effective because of its capacity to consider the vagueness and uncertainties of the decision-makers.

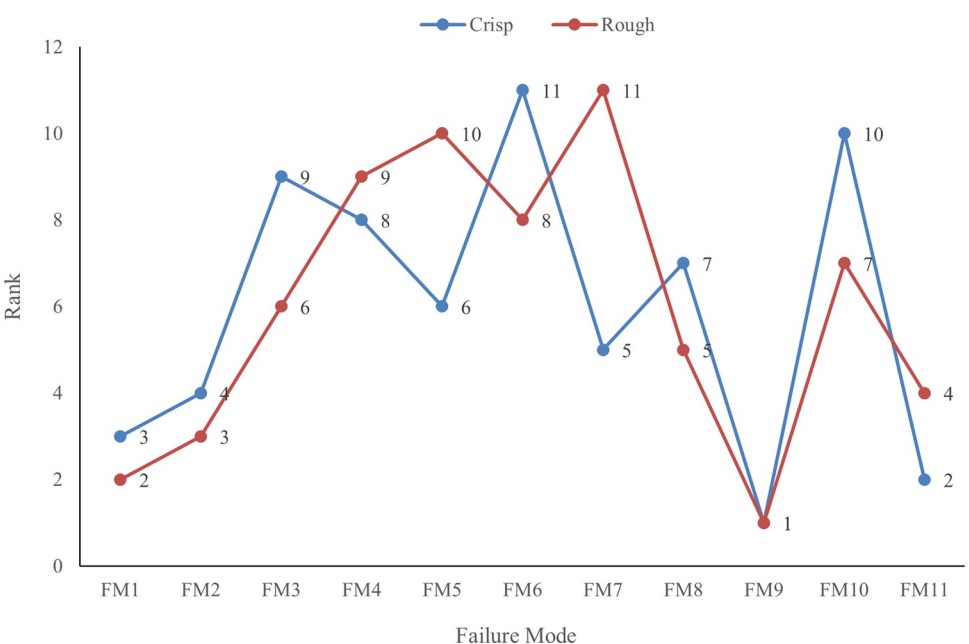

**Fig 3. Ranking of failure mode using rough TOPSIS and crisp TOPSIS methods.**

### 3.3 Discussions

From Table 7 of section 3.1, it is evident that cyber-attack, database hack risks, server failures, network interruptions, broadcast data errors, and virus effects possess the top six positions among the eleven failure factors of the IT system in the financial sector of Bangladesh.

Cyber-attacks pose a threat to the multidimensional sector, while most of the financial activities are largely dependent on the internet. Though efficient business management and automation of processes may induce operational virtue [57], cloud computing is likely to secure credentials, although makes it vulnerable to some extent. To detect and mitigate banking Trojan, a Cyber Kill Chain (CKC) based taxonomy can be implemented [58]. The software and other IT industries of Bangladesh are susceptible to such attacks as well. To protect both financial institutions from such attacks, enhanced online monitoring, usage of improved and updated firewalls, usage of stronger malware and virus protection software, etc. steps can be taken.

The database of a financial institution is considered an important asset to the organization. Human intervention and ill motives are often responsible for the security failure that jeopardizes this important asset. Although different organizations maintain their company database in their own ways, the risk of security and data loss by database hack still remains.Improved multilayer security protocol, enhanced encryption, stricter access control, etc. can be adopted to ensure database security [59,60]. Moreover, the involvement of third parties in database management can be a weak link for many financial institutions. Appointing in-house skilled IT personnel can assist to reduce this threat to a great extent.

Server failure can create a major impedance in banking operations. Such risks must be addressed tactfully to minimize SSC failure. Server failure holds the third position with a closeness coefficient value near to the value of database hack.A recent study shows an upward trend in online banking in Bangladesh, including transactions through the internet, mobile phones, ATMs, and nominated agents [61]. All these services can be severely affected if any server failure occurs. By keeping multiple backup servers, such service disruptions can be avoided.

Network interruption or link failure can also cause significant service interruption. After an evident network failure, detection and repairing strategies can often be quite time-consuming. However, with the early detection of link failure, the network failure problem can be diagnosed easily. For conspicuous improvement in reliability, modern data centers implement various proactive measures against broadcast data error. Some such notable measures include regular network maintenance, checking remote management systems, updating the operating system and control panel, checking for node redundancy, etc. These measure needs to be taken seriously to avoid future network failure.

Attacks from various viruses on the financial sector have become quite frequent these days. There is no alternative to collaborative measures on using up-to-date technology and IT audits. Cloud-based data storage is also susceptible to attacks from viruses and hacking. Suspicious e-mail, unauthorized USB usage, malicious site access, pirated software usage, etc. have been identified as prime sources of viral attacks and cybersecurity breaches. The recent investigations conducted by the Computer Incident Response Team (CIRT) of Bangladesh Bank found the presence of multiple viruses and malware in three of the Internet Service Providers (ISPs) that provide network support to multiple banks, especially when there is an alarming rate rise in the ransomware virus attacks in the Bangladeshi financial institutions [62]. The exact sources of cyber-attacks are often hard to identify as they can happen from multiple sources simultaneously [63]. Staying vigilant and adhering to all the standardized protocols, updating virus signatures, updating firewall, cleaning endpoints regularly are some of the most effective ways to thwart such attacks.

### 3.4 Managerial implications

Managers of financial institutions can be immensely benefitted from this research. Especially in developing or underdeveloped countries, where resources are constrained, it is often not possible for managers to take on multiple issues at the same time. Since this research presents and ranks the factors that contribute to the failure of information systems in the banks and other financial institutions, managers will get a clear idea about which area they should prioritize, if the resource is inadequate.

This research also highlights the preventive measures that banks can take to avoid information system failure. This is expected to make managers more aware of important issues like cybersecurity, access control, data encryption, etc. as preventive measures and help them in identification, assessment, and forecasting of future security threats. Managers of other similar multidisciplinary sectors in the developing counties can also utilize this research for evaluation and comparison of failure factors in their respective areas.

## 4 Conclusions

The financial market is growing faster than ever all around the globe. This business is no longer confined by the borders. With the development of technology, the threats took over new dimensions. Reducing various failures in the SSC is a crucial task for achieving the company's success. Managers need to recognize the failures and take proactive measures to minimize the impact of the failures. However, there hasn't been much research in this area that can assist the managers in this regard. In this research, a rough-TOPSIS based flexible FMEA model has been proposed to evaluate the SSC failures in the context of multidisciplinary sectors like banking and other similar financial industries. Existing literature review and experts' feedback helped us to identify eleven relevant SSC failure factors in this area. Subsequently, a rough-TOPSIS method was used to rank these failures. The results show that cyber-attack, database hacks, and server failures are the top three failures among the eleven failure factors.

The primary contribution of this study is the identification and evaluation of SSC failure factors of the IT system in the financial sector. This study will assist the managers in identifying the crucial factors contributing to the failures of the IT system in the banking industry and thereby, will guide the managers to minimize the effects of the failures. It will be easier for the managers to take proactive policies to reduce the number and impact of the failures in the financial sector once failure factors are properly identified and prioritized.Moreover, managers in developing countries can also utilize this research to decide, on which area they should focus first to minimize information system failure in their institutions, given that they often work with constrained resources.

The research has some limitations as well, on which future researchers can focus to overcome them. For example, maintainability is one of the risk factors that has not been considered in this study while analyzing the impacts of SSC failures. Therefore, there is a scope for further research on the impact of maintainability risks on the overall supply chain of the financial industries. Again, this study is limited by the literature review and the factors pointed out by the expert. Morediverse and multidisciplinary failure factors like changing management, failure in capacity management, etc. can also be considered in future research, without confining it to using only the feedbacks from the expert panel. Considered factors are mostly reactive types, but proactive factors could also be taken into account to improve failure response and to reduce the impact of failures. This study can also be carried out with different other MCDM methods like BWM, FUCOM, LBWA, MABAC, MAIRCA, CODAS, EDAS, etc.and the obtained results can be compared with the results of the current studyin future, to check whether the ranking or the weights of the factors change if a different approach is used.

Moreover, design flaws and impact analyses have not been carried out in the study. Lack of literature in the corresponding field of Bangladesh leaves evident gaps in this research as well.

## Supporting information

**S1 Data.**
(XLSX)

**S1 File.**
(DOCX)

## Author Contributions

**Conceptualization:** Syed Mithun Ali, S. M. Nazmul Hoq.

**Data curation:** Syed Mithun Ali, S. M. Nazmul Hoq.

**Formal analysis:** S. M. Nazmul Hoq.

**Investigation:** S. M. Nazmul Hoq, A. B. M. Mainul Bari, Sanjoy Kumar Paul.

**Methodology:** Syed Mithun Ali, S. M. Nazmul Hoq.

**Software:** Golam Kabir.

**Supervision:** Syed Mithun Ali.

**Validation:** A. B. M. Mainul Bari.

**Visualization:** Syed Mithun Ali, A. B. M. Mainul Bari, Golam Kabir, Sanjoy Kumar Paul.

**Writing – original draft:** S. M. Nazmul Hoq.

**Writing – review & editing:** Syed Mithun Ali, A. B. M. Mainul Bari, Golam Kabir, Sanjoy Kumar Paul.

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
