## [Decision Letter · Decision Letter 0]

5 Nov 2021

PONE-D-21-30749Evaluating Factors Contributing to the Failure of Information System in the Banking IndustryPLOS ONE

Dear Dr. Ali,

Thank you for submitting your manuscript to PLOS ONE. After careful consideration, we feel that it has merit but does not fully meet PLOS ONE’s publication criteria as it currently stands. Therefore, we invite you to submit a revised version of the manuscript that addresses the points raised during the review process.

We look forward to receiving your revised manuscript.

Kind regards,

Fausto Cavallaro, PhD

Academic Editor

PLOS ONE

Journal Requirements:

“NO”

4. Please ensure that you refer to Figure 1 in your text as, if accepted, production will need this reference to link the reader to the figure.

Reviewers' comments:

Reviewer's Responses to Questions

**Comments to the Author**

1. Is the manuscript technically sound, and do the data support the conclusions?

Reviewer #1: Yes

Reviewer #2: Yes

2. Has the statistical analysis been performed appropriately and rigorously? 

Reviewer #1: N/A

Reviewer #2: Yes

3. Have the authors made all data underlying the findings in their manuscript fully available?

Reviewer #1: Yes

Reviewer #2: Yes

4. Is the manuscript presented in an intelligible fashion and written in standard English?

Reviewer #1: Yes

Reviewer #2: Yes

5. Review Comments to the Author

Reviewer #1: FMEA based TOPSIS method is an effective method that is frequently used in the literature. A study applied to the banking industry has been conducted. I have some advice to the authors on the following topics:

1. By exemplifying the use of recent FMEA and TOPSIS method in different fields (especially by showing some different approaches), you should state that this method is applicable to security problems in the banking system. The following studies can help: (https://doi.org/10.1007/s00500-020-05108-y, 10.1007/s00170-014-6466-3, https://doi.org/10.1002/qre.1791 etc.)

2. In root cause trend of IT incidents we generally see changing management, 3rd party failure, software/application issues, cyber attack, hardware issues, human errors, process or control errors, failure in capacity management, some external factors etc. So, do you think your factors in table 1 are sufficient? Also, your detailed discussion of solutions to these problems will make the article more powerful.

3. I strongly suggest to the authors to read https://publications.parliament.uk/pa/cm201919/cmselect/cmtreasy/224/224.pdf

4. It would be more useful if this study was comparative. There are many different MCDM techniques available in the literature to rank these risks. I wonder what the weights would be if sorting was done with another approach?

5. If the education levels of the experts can be shared, it can give an idea and be useful in terms of how they approach the analysis. Are these IT professionals, for example, computer engineers?

6. The last paragraph of the conclusion is quite unnecessary, it is recommended to delete it.

If these minor revisions are reviewed, the article may be accepted for publication.

Reviewer #2: Review report for the paper “Evaluating Factors Contributing to the Failure of Information System in the Banking Industry”

The applicability of the method. Why do we need application rough numbers in this study? I did not see the author discussing the reason. Therefore, it is impossible to prove the superiority of this model combination in this article. Need detailed further explanation.

Insufficient expression on innovative explanations. Does the practical significance of this innovation exist? There is a lack of comparison with previous studies of the same kind. For this point, the innovativeness of the author's statement needs further explanation.

Literature review. Add more recent papers published in last three years. Remove papers published before 2017. Based on the LR you should define the scientific gap. I suggest authors to read and discuss following papers with rough stes application in MCDM field: Career selection of students using hybridized distance measure based on picture fuzzy set and rough set theory. Decision Making: Applications in Management and Engineering, 4(1), 104-126.; A novel integrated fuzzy PIPRECIA – interval rough SAW model: green supplier selection. Decision Making: Applications in Management and Engineering, 3(1), 126-145.; Sustainable supplier selection using combined FUCOM – Rough SAW model. Reports in Mechanical Engineering, 1(1), 34-43.; Parametric analysis of a grinding process using the rough sets theory. Facta universitatis series: Mechanical engineering. 18(1), 91-106. doi: 10.22190/FUME191118007A. A hybrid LBWA - IR-MAIRCA multi-criteria decision-making model for determination of constructive elements of weapons. Facta universitatis series: Mechanical Engineering, 18(3), 399-418. https://doi.org/10.22190/FUME200528033B.

Model selection problem. The author points out that it uses a MCDA method (TOPSISI) and wants to illustrate its innovation in model selection. There is no comparative proof, no analysis of the superiority of the method. Lack of comparison of results under different models. Not that the new method is equally applicable to all problems.

Criteria weights calculation. Why you have used rough FMEA method for determining criteria weights? Why not BWM, FUCOM or Level Based Weight Assessment (LBWA) methods? These methods should be discussed. The authors need to discuss their contributions compared to those in related papers. The authors must clearly discuss the significance of the research problem in the first section.

Why you have used extension of TOPSIS method? Why not MABAC, MAIRCA, CODAS, EDAS etc? These methods should be discussed. The authors need to discuss their contributions compared to those in related papers. This have to be clarified to the readers.

Rough numbers presents imprecisions in experts’ preferences, but here I can’t see experts’ individual matrices for FMEA and TOPSIS methods.

Table A1 Basic equations rough set theory and rough number – Equations for are not properly presented.

There is no result robustness. The author needs to give more detailed data references or results.

The method innovation and application value of the improved multi criteria decision model in this paper need the author to provide numerical comparison demonstration.

In the part of research status, the outline of the whole research is not clear enough, and more content of multi criteria decision model (method) needs to be added.

The results of the application part of the model need to be rearranged, the readability is too poor, and the graphical results provided can’t make people see the differences under different scene settings.

Add limitation of the method.

6. PLOS authors have the option to publish the peer review history of their article (what does this mean?). If published, this will include your full peer review and any attached files.

Reviewer #1: No

Reviewer #2: No

---

## [Author Response · Author response to Decision Letter 0]

20 Feb 2022

Evaluating Factors Contributing to the Failure of Information System in the Banking Industry

Manuscript # PONE-D-21-30749

Revision Response

We thank the reviewers for taking the time to review our paper and for their valuable comments. We thoroughly revised the paper following your suggestions and valuable feedbacks. We strongly believe that the comments, criticisms, and suggestions improved the quality of the manuscript over its earlier version. We are so pleased to resubmit the revised version for your review. We fully believe the paper will meet your requirements. The corrections and changes incorporated are being highlighted with red coloured font both in this response and in the paper for the visual convenience of the reviewers. The major changes and necessary corrections in the manuscript are detailed as follows:

Reviewer # 1

Comments:

FMEA based TOPSIS method is an effective method that is frequently used in the literature. A study applied to the banking industry has been conducted. I have some advice to the authors on the following topics:

Response: We would like to thank you for your complimentary evaluation and inspiration. Your advice and comments helped us improve the quality of the work. We have tried our best to modify the manuscript based on your comments. Please find reply to each of your comment. 

1. By exemplifying the use of recent FMEA and TOPSIS method in different fields (especially by showing some different approaches), you should state that this method is applicable to security problems in the banking system. The following studies can help: (https://doi.org/10.1007/s00500-020-05108-y, https://doi.org/10.1007/s00170-014-6466-3 , https://doi.org/10.1002/qre.1791 etc.)

Response: Thank you for your suggestion. The suggested references have been added inside the paper to justify the use of FMEA and TOPSIS method for this paper. Please check line 21-27 of Page 6 in the manuscript. Again, we are thankful for your valuable feedback to enhance the quality of our paper.

//In this study, a rough TOPSIS based FMEA approach has been used for effective identification and prioritization of the most significant failures. FMEA and TOPSIS variants have been used together before in several recent research involving failure and risk analysis. For example, Vahdani et al., (2015) utilized this approach to assessing the failure causes of steel production process, and Selim et al., (2016) developed a dynamic maintenance planning framework for an international food company. Recently, Başhan et al. (2020) used these for maritime risk evaluation and ship navigation safety. //

Reference: 

Başhan, V., Demirel, H., & Gul, M. (2020). An FMEA-based TOPSIS approach under single valued neutrosophic sets for maritime risk evaluation: the case of ship navigation safety. Soft Computing, 24(24), 18749-18764.

Selim, H., Yunusoglu, M. G., & Yılmaz Balaman, Ş. (2016). A dynamic maintenance planning framework based on fuzzy TOPSIS and FMEA: application in an international food company. Quality and Reliability Engineering International, 32(3), 795-804.

Vahdani, B., Salimi, M., & Charkhchian, M. (2015). A new FMEA method by integrating fuzzy belief structure and TOPSIS to improve risk evaluation process. The International Journal of Advanced Manufacturing Technology, 77(1-4), 357-368.//

2. In root cause trend of IT incidents we generally see changing management, 3rd party failure, software/application issues, cyber attack, hardware issues, human errors, process or control errors, failure in capacity management, some external factors etc. So, do you think your factors in table 1 are sufficient? Also, your detailed discussion of solutions to these problems will make the article more powerful.

Response: Thank you for your valuable comment. 

We would like to draw your attention towards the Table 1 (in page 10) of the paper, where you will see that, 3rd party failure (FM10), software/application issues (FM8), cyber attack (FM1), hardware issues (FM2), human errors (FM5), process or control errors (FM11,FM6), some external factors (FM3, FM7) were already included in our study (They are named in different ways, but they indicate the same issue). 

//

Table 1: Failure factors of IT System in the Banking Industry

No. Failure factors Source

FM1 Database Hack (Nakatani et al., 2018), (Mukherjee & Sengupta, 2016),(Lu & Huang, 2013)

FM2 Server failure (Randazzo et al., 2005), (Kanizo et al., 2017)

FM3 Virus Effect (Lin & Lin, 2019),(Boyson, 2014)

FM4 Cipher to Plain Text Malfunction (Khanna, 2015)

FM5 Character Misspelled Proposed in this research

FM6 Wrong Message Transcription Proposed in this research

FM7 Peripheral Error Proposed in this research

FM8 Broadcast Data Error (Up/Down) link Failure (Molero et al., 2002), (Samuels et al., 2018)

FM9 Cyber Attack (Lai et al., 2017)

FM10 Third Party Intervention (De Gusmão et al., 2016)

FM11 Network Interruption (Zhu et al., 2004), (Shiri & Akbari, 2021)

//

As for Changing management and failure in capacity management, they were not provided by the expert feedback or review of previous literatures and thus have not been included in our study. However, we sincerely mentioned this as future research in the Conclusions section. Please check line 23-26 of Page 27 in the manuscript.

//

Again, this study is limited by the literature review and the factors pointed out by the expert. More diverse and multidisciplinary failure factors like changing management, failure in capacity management, etc. can also be considered in future research, without confining it to using only the feedbacks from the expert panel.//

//

Solutions to the Top ranked failure factors has been now discussed in the newly reconstructed discussion section of the manuscript. Please see section 3.3 in page 24-26 of the paper.

//

3.3 Discussions

From Table 7 of section 3.1, it is evident that cyber-attack, database hack risks, server failures, network interruptions, broadcast data errors, and virus effects possess the top six positions among the eleven failure factors of the IT system in the financial sector of Bangladesh.

Cyber-attacks pose a threat to the multidimensional sector, while most of the financial activities are largely dependent on the internet. Though efficient business management and automation of processes may induce operational virtue (Subramani, 2012), cloud computing is likely to secure credentials, although makes it vulnerable to some extent. To detect and mitigate banking Trojan, a Cyber Kill Chain (CKC) based taxonomy can be implemented (Kiwia et al., 2018). The software and other IT industries of Bangladesh are susceptible to such attacks as well. To protect both financial institutions from such attacks, enhanced online monitoring, usage of improved and updated firewalls, usage of stronger malware and virus protection software, etc. steps can be taken.

The database of a financial institution is considered an important asset to the organization. Human intervention and ill motives are often responsible for the security failure that jeopardizes this important asset. Although different organizations maintain their company database in their own ways, the risk of security and data loss by database hack still remains. Improved multilayer security protocol, enhanced encryption, stricter access control, etc. can be adopted to ensure database security (Kamaraj, 2021; Mousa et al., 2020). Moreover, the involvement of third parties in database management can be a weak link for many financial institutions. Appointing in-house skilled IT personnel can assist to reduce this threat to a great extent.

Server failure can create a major impedance in banking operations. Such risks must be addressed tactfully to minimize SSC failure. Server failure holds the third position with a closeness coefficient value near to the value of database hack. A recent study shows an upward trend in online banking in Bangladesh, including transactions through the internet, mobile phones, ATMs, and nominated agents (Islam et al., 2019). All these services can be severely affected if any server failure occurs. By keeping multiple backup servers, such service disruptions can be avoided. 

Network interruption or link failure can also cause significant service interruption. After an evident network failure, detection and repairing strategies can often be quite time-consuming. However, with the early detection of link failure, the network failure problem can be diagnosed easily. For conspicuous improvement in reliability, modern data centers implement various proactive measures against broadcast data error. Some such notable measures include regular network maintenance, checking remote management systems, updating the operating system and control panel, checking for node redundancy, etc. These measure needs to be taken seriously to avoid future network failure.

Attacks from various viruses on the financial sector have become quite frequent these days. There is no alternative to collaborative measures on using up-to-date technology and IT audits. Cloud-based data storage is also susceptible to attacks from viruses and hacking. Suspicious e-mail, unauthorized USB usage, malicious site access, pirated software usage, etc. have been identified as prime sources of viral attacks and cybersecurity breaches. The recent investigations conducted by the Computer Incident Response Team (CIRT) of Bangladesh Bank found the presence of multiple viruses and malware in three of the Internet Service Providers (ISPs) that provide network support to multiple banks, especially when there is an alarming rate rise in the ransomware virus attacks in the Bangladeshi financial institutions (Haque & Bhuiyan, 2017). The exact sources of cyber-attacks are often hard to identify as they can happen from multiple sources simultaneously (Li et al., 2019). Staying vigilant and adhering to all the standardized protocols, updating virus signatures, updating firewall, cleaning endpoints regularly are some of the most effective ways to thwart such attacks.

//

3. I strongly suggest to the authors to read https://publications.parliament.uk/pa/cm201919/cmselect/cmtreasy/224/224.pdf

Response: Thank you for your valuable comment. We have read the suggested publication and in response we added a following lines in our paper based on it. Please check line 25-29 of Page 2 and line 1-2 of Page 3 in the manuscript.

//

In recent times, the banks and other financial industries are adopting more and more new technologies in their businesses, to streamline their operations and to gain significant advantages in the increasingly competitive market (Gupta et al., 2001; Valls Martínez et al., 2020). Consequently, there has been a drastic shift in the way that customers now access to their financial services. An increasing number of customers are now using digital or IT financial services via computers or mobile devices. As customers come to rely more heavily on these IT channels, the resilience and availability of these channels have become an important issue, since it is likely that even any brief disruption in these channels can cause significant concern among consumers (House of Commons Treasury Committee report on IT failures in the Financial Services Sector, 2019).

Added References

House of Commons Treasury Committee report on IT failures in the Financial Services Sector (2019, October 22). Retrieved from https://publications.parliament.uk/pa/cm201919/cmselect/cmtreasy/224/224.pdf

//

4. It would be more useful if this study was comparative. There are many different MCDM techniques available in the literature to rank these risks. I wonder what the weights would be if sorting was done with another approach?

Response: Thank you for your precious suggestions. As a future research scope, we can use different MCDM methods for this research and compare their results to do a comparative study. We have mentioned it in line 28-30 of Page 27 of this manuscript. We compare the weights obtained with rough TOPIS with crisp TOPIS method. The results are displayed in Figure 2 and 3. We hope these two figures and the related discussions satisfy your requirements.

5. If the education levels of the experts can be shared, it can give an idea and be useful in terms of how they approach the analysis. Are these IT professionals, for example, computer engineers?

Response: Thank you for your comment. All the experts who took part in this study are either bank officials/personnel or IT professional, who hold at least a Masters degree in their relevant area of expertise. The majority of the participating IT professionals have degree in computer engineering. However, the experts were not comfortable to share and publish their exact academic background. Therefore, this information is not included in this study. 

This has now been mentioned in the last 6 lines of Page 17, before Table 2.

//

All the experts who took part in this study are either bank officials/personnel or IT professional, who hold at least a Masters degree in their relevant area of expertise. The majority of the participating IT professionals have degree in computer engineering. However, the experts were not comfortable to share and publish their exact academic background. Therefore, this information is not included in this study. A brief summary of the experts based on their experience is listed in Table 2. 

//

6. The last paragraph of the conclusion is quite unnecessary, it is recommended to delete it.

If these minor revisions are reviewed, the article may be accepted for publication.

Response: Thank you for your precious suggestions. We have deleted the last paragraph of the conclusion as per your recommendation.

Deleted paragraph //Although this paper has only demonstrated a framework implementing integrated TOPSIS and rough set theory for IT failure assessment in the context of the banking sector, this framework can be modified to be used for failure assessment in other sectors as well, such as pharmaceuticals, health sector, telecom industry, airlines, processed food industry.//

Reviewer # 2

Comments:

1. The applicability of the method. Why do we need application rough numbers in this study? I did not see the author discussing the reason. Therefore, it is impossible to prove the superiority of this model combination in this article. Need detailed further explanation.

Response: Thank you for your valuable note. We improved the justification and advantages of using Rough TOPSIS method. The following new references were added to address this comment. Please see the Introduction section in page 6, lines 28-30 and page 7, line 1-11. 

// A rough TOPSIS method has been used here, which combines rough set theory with the traditional TOPSIS method (Yang et al., 2017). The Rough Set theory addresses the uncertainty of human judgments, where performance rating and weights cannot be assigned accurately (He et al., 2016). Hence, in this study, the framework integrates the strength of rough set theory to tackle vagueness and the merit of the TOPSIS assessment structure. It is used in most cases where the study involves dealing with imprecise or incomplete information (Božanić et al., 2020). For instance, this mehod have been used successfully for supplier selection (Đalić et al., 2020; Durmić et al., 2020), career path selection for students (Sahu et al., 2021), parametric analysis for machining process (Agarwal et al., 2020) and so on. The reason rough TOPSIS is often preferred in much recent research is that it not only improves the reliability of the TOPSIS calculation program but also express more potential information considering the uncertainities (Lo et al., 2019; Yang et al., 2017). The proposed rough TOPSIS based on flexible FMEA evaluates the failure modes except for prior information and made the execution of the FMEA process very effective (Song et al., 2014). //

Newly added References: 

Agarwal, S., Dandge, S. S., & Chakraborty, S. (2020). PARAMETRIC ANALYSIS OF A GRINDING PROCESS USING THE ROUGH SETS THEORY. Facta Universitatis, Series: Mechanical Engineering, 18(1), 091-106.

Božanić, D., Ranđelović, A., Radovanović, M., & Tešić, D. (2020). A hybrid LBWA-IR-MAIRCA multi-criteria decision-making model for determination of constructive elements of weapons. Facta Universitatis, Series: Mechanical Engineering, 18(3), 399-418.

Đalić, I., Stević, Ž., Karamasa, C., & Puška, A. (2020). A novel integrated fuzzy PIPRECIA–interval rough SAW model: Green supplier selection. Decision Making: Applications in Management and Engineering, 3(1), 126-145.

Durmić, E., Stević, Ž., Chatterjee, P., Vasiljević, M., & Tomašević, M. (2020). Sustainable supplier selection using combined FUCOM–Rough SAW model. Reports in mechanical engineering, 1(1), 34-43.

Sahu, R., Dash, S. R., & Das, S. (2021). Career selection of students using hybridized distance measure based on picture fuzzy set and rough set theory. Decision Making: Applications in Management and Engineering, 4(1), 104-126.

//

Moreover, a new section (2.3) with some new discussions have been added to discuss rough theory, following your suggestion. Please see line 1-10 of Page 13 of the manuscript.

//

2.3 Rough Set Theory

Rough set theory has applications in many areas of research. One of the most important application of rough set theory is for elimination of impact of the vagueness in the decision making (He et al., 2016). For example, in the area of decision analysis, the decision-makers are required to evaluate the criteria for a particular problem and provide the feedback on them using some particular scaled values. Since it is not always possible to make sure that all the decision-makers are experts in all fields, an unexperienced decision-maker can decide on a particular area and the judgment made by that expert might contain uncertainty. In order to find and eliminate these uncertainties, rough set theory plays an important role (Song et al., 2014). Basic equations of rough set theory and rough number are presented in the Appendix. 

//

2. Insufficient expression on innovative explanations. Does the practical significance of this innovation exist? There is a lack of comparison with previous studies of the same kind. For this point, the innovativeness of the author's statement needs further explanation.

Response: Thank you for your valuable comment. 

Practical significance of this innovation does exist. In this new age of technology, customers are increasingly being expected to use digital services, and yet these services are being significantly disrupted due to IT failures. Consumers suffer from various issues when these IT failures occur. 

We feel sorry that we did not compare our findings with similar previous studies. Although we conceptualized the failure factors based on previous studies, we failed to find previous studies of the same kind where ranking of IT failure factors in the banking industry was investigated under the lens of a multicriteria decision making approach. Therefore, a research gap does exist, and we attempted to fill the gap. To justify the innovation discussed in this paper, following sentences has been added to the paper. Please check line 25-29 of Page 2 and line 1-2 of Page 3 in the manuscript.

//In recent times, the banks and other financial industries are adopting more and more new technologies in their businesses, to streamline their operations and to gain significant advantages in the increasingly competitive market (Gupta et al., 2001; Valls Martínez et al., 2020). Consequently, there has been a drastic shift in the way that modern customers now access their financial services. An increasing number of customers are now using digital or IT financial services via computers or mobile devices. As customers come to rely more heavily on these IT channels, the resilience and availability of these channels is has become an important issue, since it is likely that even any brief disruption in these channels can cause significant concern among consumers (House of Commons Treasury Committee report on IT failures in the Financial Services Sector, 2019).

References

House of Commons Treasury Committee report on IT failures in the Financial Services Sector ,2019, October 22. Retrieved from https://publications.parliament.uk/pa/cm201919/cmselect/cmtreasy/224/224.pdf

//

To make the research contribution clearer, we have rephrased and modified subsection 3.3 (Discussion) and section 4 (Conclusion). Please see Page 24-28 in the manuscript.

More details on the managerial/research implication can be found in the newly created subsection 3.3 (Managerial Implications) in line 11-23 of Page 26 in the manuscript.

//

3.3 Managerial implications 

Managers of the financial institutions can be immensely benefitted from this research. Specially in the developing or underdeveloped countries, where resources are constrained, it is often not possible for the managers to take on multiple issues at the same time. Since this research presents and ranks the factors that contribute to the failure of information system in the banks and other financial institutions, managers will get a clear idea about which area they should prioritize, if the resource is inadequate. 

This research also highlights on the preventive measures that banks can take to avoid information system failure. This is expected to make mangers more aware on important issues like cybersecurity, access control, data encryption, etc. as preventive measures and help them in identification, assessment, and forecasting of future security threats. Managers of other similar multidisciplinary sectors in the developing counties can also utilize this research for evaluation and comparison of failure factors in their respective areas.

//

3. Literature review. Add more recent papers published in last three years. Remove papers published before 2017. Based on the LR you should define the scientific gap. I suggest authors to read and discuss following papers with rough sets application in MCDM field: 

 Career selection of students using hybridized distance measure based on picture fuzzy set and rough set theory. Decision Making: Applications in Management and Engineering, 4(1), 104-126.; 

 A novel integrated fuzzy PIPRECIA – interval rough SAW model: green supplier selection. Decision Making: Applications in Management and Engineering, 3(1), 126-145.; 

 Sustainable supplier selection using combined FUCOM – Rough SAW model. Reports in Mechanical Engineering, 1(1), 34-43.; 

 Parametric analysis of a grinding process using the rough sets theory. Facta universitatis series: Mechanical engineering. 18(1), 91-106. doi: 10.22190/FUME191118007A. 

 A hybrid LBWA - IR-MAIRCA multi-criteria decision-making model for determination of constructive elements of weapons. Facta universitatis series: Mechanical Engineering, 18(3), 399-418. https://doi.org/10.22190/FUME200528033B.

Response: Thank you for your valuable comment. We feel sorry that it is not possible to eliminate all papers published before 2017, since many of them were very relevant and those were cited for theoretical development. However, we have added all the suggested references to improve our argument to use rough set theory in our paper. Many new recent references have also been added in relevant places during this review phase. Please see lines 28-30 of page 6 and lines 1-11 of Page 7 of this manuscript.

////The Rough Set theory addresses the uncertainty of human judgments, where performance rating and weights cannot be assigned accurately (He et al., 2016). It is used in most cases where the study involves dealing with imprecise or incomplete information (Božanić et al., 2020). For instance, this method have been used successfully for supplier selection (Đalić et al., 2020; Durmić et al., 2020), career path selection for students (Sahu et al., 2021), parametric analysis for machining process (Agarwal et al., 2020) and so on.///

Added Reference: 

Agarwal, S., Dandge, S. S., & Chakraborty, S. (2020). PARAMETRIC ANALYSIS OF A GRINDING PROCESS USING THE ROUGH SETS THEORY. Facta Universitatis, Series: Mechanical Engineering, 18(1), 091-106.

Božanić, D., Ranđelović, A., Radovanović, M., & Tešić, D. (2020). A hybrid LBWA-IR-MAIRCA multi-criteria decision-making model for determination of constructive elements of weapons. Facta Universitatis, Series: Mechanical Engineering, 18(3), 399-418.

Đalić, I., Stević, Ž., Karamasa, C., & Puška, A. (2020). A novel integrated fuzzy PIPRECIA–interval rough SAW model: Green supplier selection. Decision Making: Applications in Management and Engineering, 3(1), 126-145.

Durmić, E., Stević, Ž., Chatterjee, P., Vasiljević, M., & Tomašević, M. (2020). Sustainable supplier selection using combined FUCOM–Rough SAW model. Reports in mechanical engineering, 1(1), 34-43.

Sahu, R., Dash, S. R., & Das, S. (2021). Career selection of students using hybridized distance measure based on picture fuzzy set and rough set theory. Decision Making: Applications in Management and Engineering, 4(1), 104-126.

//

As for scientific/research gap, several new studies have been added to identify the research gaps. However, since there has not been much recent research in this area, the newly cited papers are not very recent. Please see the lines 29-30 of Page 3 and line 1- 11 of Page 4 in the manuscript for more details.

//

However, it has been observed that, even though the rest of the world is well aware of the safety and security of IT-based banking, the banking sector, especially in Bangladesh, is still struggling with it. Although technology being a propelling factor of the economy, there exist threats and failures to safeguard the business from various existing loopholes (Smerlak et al., 2014). Clementina and Isu (2016) evaluated the insecure situation, bank fraud and their impact on bank performance in perspective of the commercial banks of Nigeria. The study used a multiple regression analysis to determine if there is any significant relationship between the indicators of bank insecurity and fraud. Ula et al. (2011) explores the relation between the information assets and potential threats for banking system. The study also examines and compares the elements from the commonly used information security governance frameworks, standards and best practices. Edge et al. (2007) tried to help the banks and other financial institutions to identify how attackers compromise accounts and develop methods to protect them. They used an ‘attack trees and protection trees’ methods to do this. Thereby, it is evident that there has not been much research on the identification and analysis of the factors contributing to the IT failures in the in financial institution, in the previous years, which presents a clear research gap.

 Hence, this research intends to shed light on the factors that contribute to the failure of the banking IT systems. After identification of the factors contributing to the failure of the IT systems in the banking industry, this research proposes a rough-TOPSIS (Technique for Order of Preference by Similarity to Ideal Solution) based flexible Failure Mode and Effect Analysis (FMEA) approach to evaluate the failure factors. //

4. Model selection problem. The author points out that it uses a MCDA method (TOPSIS) and wants to illustrate its innovation in model selection. There is no comparative proof, no analysis of the superiority of the method. Lack of comparison of results under different models. Not that the new method is equally applicable to all problems.

Response: Thank you for your comment. We have added some justifications for using rough TOPSIS over crisp TOPSIS in the newly added section 3.3 of the paper following your suggestion. Please see Page 22-24 in the manuscript for more details.

//3.2 Model Comparison

Figure 2 presents the graphical representation of a comparison of weights of severity, occurrence, detection, time, and cost obtained by the rough method to the conventional crisp method. It is noteworthy to mention that the order of ranking of weights of all the factors by the rough method is almost the same as the rank order of the crisp method. Occurrence > Detection > Severity > Time > Cost is the rank order obtained by rough method while the sequence of rank by crisp method is Occurrence > Severity > Detection > Time > Cost.

Figure 2: Comparison of weights using rough and crisp value

However, the rough method is effective to represent the uncertainties as it fits the values of decision-makers in the form of upper and lower limits. According to Figure 2, the spread of judgment by the experts is represented in the form of the bar for the rough method process as opposed to a line by the crisp method. The less the length of the bar indicates, the less the uncertainties of decisions by the decision-makers. The more the length of the spread represents, the lower the accuracy of the decisions. When it comes to the weights by the traditional crisp method or other MCDM methods like analytic hierarchy process (AHP), best-worst method (BWM), they are represented by a single crisp value or in the form of lines shown in Figure 2, although multiple decision-makers were involved in this decision making. All these methods consider only the mean decision value by the experts and the vagueness and uncertainties of the judgment values cannot be represented properly by these methods. 

The rank of the failure modes found by the rough TOPSIS method is also compared with the rank of failure modes by the crisp TOPSIS method and presented in Figure 3. It can be seen from the graph that cyber attack is the most critical failure factor based on both methods. The ranking of database hack, server failure, and network interruption are most similar for both methods. There exists slight ranking variation for the cipher to plain text malfunction and broadcast data error factors. However, peripheral error and character misspelled factors show the most significant difference. It ranked fifth based on the crisp TOPSIS method while eleventh based on the rough TOPSIS method. Similarly, character misspelled ranked sixth and tenth based on the crisp TOPSIS method and rough TOPSIS method, respectively. According to the crisp TOPSIS method, wrong message transcription is the least critical factor whereas the rough TOPSIS method indicates the peripheral error. The results provided by the rough TOPSIS method are more reliable and effective because of its capacity to consider the vagueness and uncertainties of the decision-makers.

Figure 3: Model comparison of weights using rough TOPSIS and crisp TOPSIS methods

//

5. Criteria weights calculation. Why you have used rough FMEA method for determining criteria weights? Why not BWM, FUCOM or Level Based Weight Assessment (LBWA) methods? These methods should be discussed. The authors need to discuss their contributions compared to those in related papers. The authors must clearly discuss the significance of the research problem in the first section.

Response: Thank you for your precious suggestions. We feel that every MCDM method has its advantages and disadvantages. We used rough FMEA over traditional FMEA for determining the criteria weight because rough FMEA offers some benefits. For instance, incorporation of rough set with FMEA eliminates the impact of vagueness in decision making. For example, in the area of decision analysis, the decision-makers are required to evaluate the criteria for a particular problem and provide feedback on them using some particular scaled values. Since it is not always possible to make sure that all the decision-makers are experts in all fields, an inexperienced decision-maker can decide on a particular area and the judgment made by that expert might contain uncertainty. Rough set theory helps to eliminate these uncertainties. This has been also discussed in the newly added section 2.3 on rough set theory in the manuscript (lines 1-10 of Page 13).

We feel sorry that we can’t incorporate BWM, FUCOM or Level Based Weight Assessment (LBWA) methods at this stage for evaluating the criteria weights. We already completed our research project and communicated the findings with our university. We are afraid recalculating the weights using these methods may jeopardize the credibility of the research. However, we sincerely incorporated your suggestion as a future research direction. See the last 3 lines of Page 27, in the conclusion section. 

//

This study can also be carried out with different other MCDM methods like BWM, FUCOM, LBWA, MABAC, MAIRCA, CODAS, EDAS, etc. and the obtained results can be compared with the results of the current study in future, to check whether the ranking or the weights of the factors change if a different approach is used.

We feel sorry that we did not compare our findings with similar previous studies. Although we conceptualized the failure factors based on previous studies, we failed to find previous studies of the same kind where ranking of IT failure factors in the banking industry was investigated under the lens of a multicriteria decision making approach. Therefore, a research gap does exist, and we attempted to fill the gap.

//

6. Why you have used extension of TOPSIS method? Why not MABAC, MAIRCA, CODAS, EDAS etc? These methods should be discussed. The authors need to discuss their contributions compared to those in related papers. This have to be clarified to the readers.

Response: Thank you for your precious suggestions. Rough TOPSIS provides some added benefits over traditional TOPSIS. The advantage of incorporating rough set theory has been now discussed in the newly added section 2.3 in the manuscript (lines 1-10 of Page 13).

//Incorporation of rough set eliminates the impact of vagueness in decision making. For example, in the area of decision analysis, the decision-makers are required to evaluate the criteria for a particular problem and provide feedback on them using some particular scaled values. Since it is not always possible to make sure that all the decision-makers are experts in all fields, an inexperienced decision-maker can decide on a particular area and the judgment made by that expert might contain uncertainty. Rough set theory helps to eliminate these uncertainties.//

Sadly, we don’t find any relevant paper that applied MABAC, MAIRCA, CODAS, EDAS for evaluating IT failure factors. Hence, we apologize that we have not discussed those methods in detail in the manuscript. In the revised manuscript, we theoretically enriched our arguments based on recent relevant articles and the articles suggested by you, all the respected reviewers. Hope, the revised article now satisfies your requirements. The following sentences are inserted in the manuscript. 

// Various MCDM techniques have been used in the area of failure and risk analysis in recent time. For example, Bathrinath et al. (2021) analyzed the risks in the textile industry using an Analytic Hierarchy Process (AHP)- Technique for Order of Preference by Similarity to Ideal Solution (TOPSIS) hybrid method. Şenel et al. (2018) analyzed the risks in the maritime industries of Turkey using FMEA based intuitionistic Fuzzy TOPSIS Approach. Pamučar et al. (2018) used a multi-criteria Full Consistency Method (FUCOM)-Multi-Attributive Ideal-Real Comparative Analysis (MAIRCA) model for the evaluation of level crossings in the Republic of Serbia. Stević and Brković (2020) utilized a hybrid FUCOM- Measurement of alternatives and ranking according to compromise solution (MARCOS) model for evaluation of human resources in a transport company. Jokić et al. (2021) used a Level Based Weight Assessment (LBWA)-Fuzzy Multi-Attributive Border Approximation area Comparison (MABAC) method for the selection of appropriate firing positions for the mortars used by the military artillery unit. Liu et al. (2020) used an integrated Stepwise Weight Assessment Ratio Analysis (SWARA)-MABAC method to assess occupational health and safety risk. Hou et al. (2021) analyzed the safety risks in the metro construction under epistemic uncertainty, using credal networks and the Evaluation Based on Distance from Average Solution (EDAS) method. Bakhat and Rajaa (2020) analyzed the risks in a wind turbine operation in Morocco using a Gray AHP-MABAC approach. Xu (2021) performed a performance evaluation in the investment environment of blockchain industry using a Fuzzy Combinative Distance based ASsesment (CODAS) method. However, there has not been any significant research using any MCDM technique on the identification and analysis of the factors contributing to the IT failures in the in financial institution so far, which presents a clear research gap.//

We cordially take the opportunity to suggest these methods as future research directions. Please See the last 3 lines of page 27, in the conclusions section.

To theoretically enriched our arguments of our proposed method, based on recent relevant articles, following line has been added and revised in line 22-31 of page 6 and lines 1- 11 of page 7. 

//In this study, a rough TOPSIS based FMEA approach has been used for effective identification and prioritization of the most significant failures. FMEA and TOPSIS variants have been used together before in several recent studies involving failure and risk analysis. For example, Vahdani et al., (2015) utilized this approach to assess the failure causes of the steel production process; and Selim et al., (2016) developed a dynamic maintenance planning framework for an international food company. Recently, Başhan et al. (2020) used these for maritime risk evaluation and ship navigation safety. 

A rough TOPSIS method has been used here, which combines rough set theory with the traditional TOPSIS method (Yang et al., 2017). The Rough Set theory addresses the uncertainty of human judgments, where performance rating and weights cannot be assigned accurately (He et al., 2016). Hence, in this study, the framework integrates the strength of rough set theory to tackle vagueness and the merit of the TOPSIS assessment structure. It is used in most cases where the study involves dealing with imprecise or incomplete information (Božanićet al., 2020). For instance, this method has been used successfully for supplier selection (Đalić et al., 2020; Durmić et al., 2020), career path selection for students (Sahu et al., 2021), parametric analysis for the machining process (Agarwal et al., 2020) and so on. The reason rough TOPSIS is often preferred in much recent research is that it not only improves the reliability of the TOPSIS calculation program but also expresses more potential information considering the uncertainties(Lo et al., 2019;Yang et al., 2017). The proposed rough TOPSIS based on flexible FMEA evaluates the failure modes except for prior information and made the execution of the FMEA process very effective (Song et al., 2014).//

7. Rough numbers presents imprecisions in experts’ preferences, but here I can’t see experts’ individual matrices for FMEA and TOPSIS methods.

Response: Thank you for your precious suggestions. We have considered total 32 decision makers in this study. However, due to the space limitations, it is not possible to provide all the matrices inside the paper. Thereby, as samples, we have included the metrices from two different decision makers in the appendix. Please see at the Appendix B (Page 37-39) of the manuscript for details.

8. Table A1 Basic equations rough set theory and rough number – Equations for are not properly presented.

Response: Thank you for your precious suggestions. We have eliminated that table in Appendix A and replaced it with elaborated description of the equation of rough set theory. Please see the revised Appendix A in Page 34-35 of the manuscript.

//

APPENDIX A

Basic equations of rough set theory and rough number (Song et al., 2014) are given below.

Considering there are n classes of experts’ opinion, R={C1,C2,…,Cn}, which are in the order C1<C2<⋯<Cn, and Y is an arbitrary object of U, then the upper and lower approximations of C_i and the boundary region are evaluated by,

Lower approximation:

▁Apr (C_i )=U{Y∈ U/R(Y)≤C_i } (1)

Upper approximation:

(Apr) ®(C_i )=U{Y∈ U/R(Y)≥C_i } (2)

Boundary region:

Bnd(C_i )=U{Y∈ U/R(Y)≠C_i }

 ={Y∈ U/R(Y)>C_i }∪{Y∈ U/R(Y)<C_i } (3)

Hence, the class C_i can be represented in the form of a rough number, which contains the lower limit ▁Lim (C_i ) and upper limit (Lim) ®(C_i ) and can be calculated as,

▁Lim (C_i )=1/N_L ∑▒〖R(Y)|Y∈〗 ▁Apr (C_i ) (4)

(Lim) ®(C_i )=1/N_U ∑▒〖R(Y)|Y∈〗 (Apr) ®(C_i ) (5)

where, N_L represents number of objects included for lower approximation of C_i, and N_U is the number of objects included for the upper approximation of C_i.

The experts’ subjective decisions can be expressed in terms of rough interval form on the basis of lower limit ▁Lim (C_i ) and upper limit (Lim) ®(C_i ).

Rough number:

RN(C_i )=[¯Lim (C_i ),▁Lim (C_i )]

(6)

The degree of accuracy of decisions by decision-makers can be analyzed by finding the interval of boundary region, and the smaller the interval of a rough number, the greater the precision is.

 Interval of boundary region:

IBR(C_i )=¯Lim (C_i )-▁Lim (C_i ) (7)

The arithmetic operations for rough numbers are done as follows:

Addition of rough numbers 〖RN〗_1 and 〖RN〗_2,

〖RN〗_1+〖RN〗_2=(▁Lim_1,¯Lim_1 )+(▁Lim_2,¯Lim_2 )=(▁Lim_1+▁Lim_2,¯Lim_1+¯Lim_2 ) (8)

Subtraction of rough numbers 〖RN〗_1 and 〖RN〗_2,

〖RN〗_1-〖RN〗_2=(▁Lim_1,¯Lim_1 )-(▁Lim_2,¯Lim_2 )=(▁Lim_1-▁Lim_2,¯Lim_1-¯Lim_2 ) (9)

Multiplication of rough numbers 〖RN〗_1 and 〖RN〗_2,

〖RN〗_1×〖RN〗_2=(▁Lim_1,¯Lim_1 )×(▁Lim_2,¯Lim_2 )=(▁Lim_1×▁Lim_2,¯Lim_1×¯Lim_2 ) (10)

Division of rough numbers 〖RN〗_1 and 〖RN〗_2,

〖RN〗_1÷〖RN〗_2=(▁Lim_1,¯Lim_1 )÷(▁Lim_2,¯Lim_2 )=(▁Lim_1÷¯Lim_2,▁Lim_2÷¯Lim_1 )

(11)

Scalar multiplication of rough number 〖RN〗_1 with non-zero constant k,

k×〖RN〗_1=〖k×▁Lim〗_1,k×¯Lim_1

(12)

//

9. There is no result robustness. The author needs to give more detailed data references or results.

Response: Thank you for your precious suggestions. We compare the results (Refer to the responses to comment 4) with another similar approach. Hope it gives the result robustness. 

Also, we made the following statement at the end of the conclusion section. Hope the statement helps audiences understand the data used in the model.

//Data Availability Statement: Data used in the model building are found in the paper. Also, An Excel file containing the raw data and the calculations has been supplied.//

10. The method innovation and application value of the improved multi criteria decision model in this paper need the author to provide numerical comparison demonstration.

Response: Thank you for your precious suggestions. As a future research scope, we are taking it under advisement to use other different MCDM methods for this research and compare their results together to carry out a comparative study. We have mentioned this in line 28-30 of Page 27 of this manuscript.

// This study can also be carried out with different other MCDM methods and the obtained results can be compared the results of the current study in future, to check whether the ranking or the weights of the factors change if a different approach is used. //

11. In the part of research status, the outline of the whole research is not clear enough, and more content of multi criteria decision model (method) needs to be added.

Response: Thank you for your precious suggestions. We have added some new discussions on the use of different MCDM methods in recent years and added some other new discussions to improve the research outline. Please check the Page 3 of the manuscript for details.

//However, it has been observed that, even though the rest of the world is well aware of the safety and security of IT-based banking,the banking sector,especially in Bangladesh, is still struggling with it. Although technology being a propelling factor of the economy, there exist threats and failures to safeguard the business from various existing loopholes (Smerlak et al., 2014). Clementina and Isu (2016) evaluates the insecure situation, bank fraud and their impact on bank performance in perspective of the commercial banks of Nigeria. The study used a multiple regression analysis to determine if there is any significant relationship between the indicators of bank insecurity and fraud. Ulaet al. (2011) explores the relation between the information assets and potential threats for banking system. The study also examines and compares the elements from the commonly used information security governance frameworks, standards and best practices. Edge et al. (2007) tried to help the banks and other financial institutions to identify how attackers compromise accounts and develop methods to protect them. They used an ‘attack trees and protection trees’ methods to do this. 

 Various MCDM techniques have been used in the area of failure and risk analysis in recent time. For example, Bathrinath et al. (2021) analyzed the risks in the textile industry using an Analytic Hierarchy Process (AHP)- Technique for Order of Preference by Similarity to Ideal Solution (TOPSIS) hybrid method. Şenel et al. (2018) analyzed the risks in the maritime industries of Turkey using FMEA based intuitionistic Fuzzy TOPSIS Approach. Pamučar et al. (2018) used a multi-criteria Full Consistency Method (FUCOM)- Multi-Attributive Ideal-Real Comparative Analysis (MAIRCA) model for the evaluation of level crossings in the Republic of Serbia. Stević and Brković (2020) utilized a hybrid FUCOM- Measurement of alternatives and ranking according to compromise solution (MARCOS) model for evaluation of human resources in a transport company. Jokić et al. (2021) used a Level Based Weight Assessment (LBWA) -Fuzzy Multi-Attributive Border Approximation area Comparison (MABAC) method for the selection of appropriate firing positions for the mortars used by the military artillery unit. Liu et al. (2020) used an integrated Stepwise Weight Assessment Ratio Analysis (SWARA)- MABAC method to assess occupational health and safety risk. Hou et al. (2021) analyzed the safety risks in the metro construction under epistemic uncertainty, using credal networks and the Evaluation Based on Distance from Average Solution (EDAS) method. Bakhat and Rajaa (2020) analyzed the risks in a wind turbine operation in Morocco using a Gray AHP-MABAC approach. Xu (2021) performed a performance evaluation in the investment environment of blockchain industry using a Fuzzy Combinative Distance based ASsesment (CODAS) method. However, there has not been any significant research using any MCDM technique on the identification and analysis of the factors contributing to the IT failures in the in financial institution so far, which presents a clear research gap.

 Hence, this research, at first, intends identify the factors that contribute to the failure of the banking IT systems from expert feedbacks and previous relevant literatures. After that, it proposes a rough-TOPSIS (Technique for Order of Preference by Similarity to Ideal Solution) based flexible Failure Mode and Effect Analysis (FMEA) approach to evaluate the identified factors. 

Newly added references here:

Bathrinath S, Bhalaji RK, Saravanasankar S. Risk analysis in textile industries using AHP-TOPSIS. Materials Today: Proceedings. 2021 Jan 1;45:1257-63.

Şenel M, Şenel B, Havle CA. Risk analysis of ports in Maritime Industry in Turkey using FMEA based intuitionistic Fuzzy TOPSIS Approach. InITM Web of Conferences 2018 (Vol. 22, p. 01018). EDP Sciences.

Pamučar D, Lukovac V, Božanić D, Komazec N. Multi-criteria FUCOM-MAIRCA model for the evaluation of level crossings: case study in the Republic of Serbia. Operational Research in Engineering Sciences: Theory and Applications. 2018 Dec 19;1(1):108-29.

Stević Ž, Brković N. A novel integrated FUCOM-MARCOS model for evaluation of human resources in a transport company. Logistics. 2020 Mar;4(1):4. 

Jokić Ž, Božanić D, Pamučar D. Selection of fire position of mortar units using LBWA and Fuzzy MABAC model. Operational Research in Engineering Sciences: Theory and Applications. 2021 Mar 28;4(1):115-35.

Liu R, Hou LX, Liu HC, Lin W. Occupational health and safety risk assessment using an integrated SWARA-MABAC model under bipolar fuzzy environment. Computational and Applied Mathematics. 2020 Dec;39(4):1-7. 

Bakhat R, Rajaa M. Risk Assessment of a Wind Turbine Using an AHP-MABAC Approach with Grey System Theory: A Case Study of Morocco. Mathematical Problems in Engineering. 2020 Aug 13;2020.

Xu Y. Research on Investment Environment Performance Evaluation of Blockchain Industry with Intuitionistic Fuzzy CODAS Method. Scientific Programming. 2021 Nov 22;2021.

Hou WH, Wang XK, Zhang HY, Wang JQ, Li L. Safety risk assessment of metro construction under epistemic uncertainty: An integrated framework using credal networks and the EDAS method. Applied Soft Computing. 2021 Sep 1;108:107436.//

To further theoretically enriched our arguments of our proposed method, based on recent relevant articles, following line has been added and revised in line 22-31 of page 6 and lines 1- 11 of page 7. 

//In this study, a rough TOPSIS based FMEA approach has been used for effective identification and prioritization of the most significant failures. FMEA and TOPSIS variants have been used together before in several recent studies involving failure and risk analysis. For example, Vahdani et al., (2015) utilized this approach to assess the failure causes of the steel production process; and Selim et al., (2016) developed a dynamic maintenance planning framework for an international food company. Recently, Başhan et al. (2020) used these for maritime risk evaluation and ship navigation safety. 

A rough TOPSIS method has been used here, which combines rough set theory with the traditional TOPSIS method (Yang et al., 2017). The Rough Set theory addresses the uncertainty of human judgments, where performance rating and weights cannot be assigned accurately (He et al., 2016). Hence, in this study, the framework integrates the strength of rough set theory to tackle vagueness and the merit of the TOPSIS assessment structure. It is used in most cases where the study involves dealing with imprecise or incomplete information (Božanićet al., 2020). For instance, this method has been used successfully for supplier selection (Đalić et al., 2020; Durmić et al., 2020), career path selection for students (Sahu et al., 2021), parametric analysis for the machining process (Agarwal et al., 2020) and so on. The reason rough TOPSIS is often preferred in much recent research is that it not only improves the reliability of the TOPSIS calculation program but also expresses more potential information considering the uncertainties(Lo et al., 2019;Yang et al., 2017). The proposed rough TOPSIS based on flexible FMEA evaluates the failure modes except for prior information and made the execution of the FMEA process very effective (Song et al., 2014).//

12. The results of the application part of the model need to be rearranged, the readability is too poor, and the graphical results provided can’t make people see the differences under different scene settings.

Response: Thank you for the suggestions. We feel sorry that we don’t understand these suggestions completely. 

We graphically compare the results (Refer to the responses to comment 4) with another similar approach. Hope it gives the result robustness. 

13. Add limitation of the method.

Response: Thank you for your precious suggestions. Several limitations of this study have been included in line 19-30 of Page 27 and line 1-2 of Page 28 in the manuscript.

//

 The research has some limitations as well, on which future researchers can focus to overcome them. For example, maintainability is one of the risk factors that has not been considered in this study while analyzing the impacts of SSC failures. Therefore, there is a scope for further research on the impact of maintainability risks on the overall supply chain of the financial industries. Again, this study is limited by the literature review and the factors pointed out by the expert. More diverse and multidisciplinary failure factors like changing management, failure in capacity management, etc. can also be considered in future research, without confining it to using only the feedbacks from the expert panel. Considered factors are mostly reactive types, but proactive factors could also be taken into account improve failure response and reduce the impact of failures. This study can also be carried out with different other MCDM methods and the obtained results can be compared with the results of the current study in future, to check whether the ranking or the weights of the factors change if a different approach is used. Moreover, design flaws and impact analyses have not been carried out in the study. Lack of literature in the corresponding field of Bangladesh leaves evident gaps in this research as well.

//

---

## [Decision Letter · Decision Letter 1]

7 Mar 2022

Evaluating Factors Contributing to the Failure of Information System in the Banking Industry

PONE-D-21-30749R1

Dear Dr. Ali,

We’re pleased to inform you that your manuscript has been judged scientifically suitable for publication and will be formally accepted for publication once it meets all outstanding technical requirements.

Kind regards,

Fausto Cavallaro, PhD

Academic Editor

PLOS ONE

**Comments to the Author**

1. If the authors have adequately addressed your comments raised in a previous round of review and you feel that this manuscript is now acceptable for publication, you may indicate that here to bypass the “Comments to the Author” section, enter your conflict of interest statement in the “Confidential to Editor” section, and submit your "Accept" recommendation.

Reviewer #1: All comments have been addressed

Reviewer #2: All comments have been addressed

2. Is the manuscript technically sound, and do the data support the conclusions?

Reviewer #1: Yes

Reviewer #2: Yes

3. Has the statistical analysis been performed appropriately and rigorously? 

Reviewer #1: N/A

Reviewer #2: Yes

4. Have the authors made all data underlying the findings in their manuscript fully available?

Reviewer #1: Yes

Reviewer #2: Yes

5. Is the manuscript presented in an intelligible fashion and written in standard English?

Reviewer #1: Yes

Reviewer #2: Yes

6. Review Comments to the Author

Reviewer #1: Article is now much better thank you for your revisions. Reference list should be corrected there are some mis information there.

Reviewer #2: The authors have addressed the point of my concern. I am happy with their corrections. Hence, I would like to recommend this manuscript to be published.

7. PLOS authors have the option to publish the peer review history of their article (what does this mean?). If published, this will include your full peer review and any attached files.

Reviewer #1: No

Reviewer #2: No

---

## [Editor Report · Acceptance letter]

9 Mar 2022

PONE-D-21-30749R1 

Evaluating Factors  Contributing to the Failure of Information System in the Banking Industry 

Dear Dr. Ali:

I'm pleased to inform you that your manuscript has been deemed suitable for publication in PLOS ONE. Congratulations! Your manuscript is now with our production department. 

Kind regards, 

on behalf of

Professor Fausto Cavallaro 

Academic Editor

PLOS ONE